# Multiphysics Bench: Benchmarking and Investigating Scientific Machine Learning for Multiphysics PDEs

## Abstract

Solving partial differential equations (PDEs) with machine learning has attracted great attention, as PDEs are fundamental tools for modeling real-world systems that range from fundamental physical science to advanced engineering disciplines. Most real-world physical systems across various disciplines are involved in multiple coupled physical fields rather than a single field. For example, in 3D integrated circuits (ICs), electrical current injection or electromagnetic wave propagation can induce localized heating, which in turn alters the electromagnetic properties of the embedded components. However, previous machine learning studies mainly focused on solving single-field problems, but overlooked the importance and characteristics of multiphysics problems in real world. Multiphysics PDEs typically entail multiple strongly coupled field quantities, thereby introducing additional complexity and challenges, such as inter-field coupling. Nevertheless, benchmark testing for the application of machine learning in solving multiphysics problems remains largely unexamined. To identify and address the emerging challenges in multiphysics problems, we make three main contributions in this work. First, we collect the first general multiphysics dataset, the Multiphysics Bench, which focuses on multiphysics PDE solving with machine learning. Multiphysics Bench is also the most comprehensive multiphysics PDE dataset to date, featuring the broadest range of coupling types, the greatest diversity of multiphysics PDE formulations, and the largest scale of coupled physics data. Second, we conduct the first systematic investigation on multiple representative learning-based PDE solvers, such as Physics-Informed Neural Networks (PINNs), Fourier Neural Operators (FNO), Deep Operator Networks (DeepONet), and DiffusionPDE solvers, on multiphysics problems. Unfortunately, naively applying these existing solvers usually shows very poor performance for solving multiphysics. Third, through extensive experiments and discussions, we report multiple insights and a bag of useful tricks for solving multiphysics with machine learning, motivating future directions in the study and simulation of complex, coupled physical systems. Notably, our multiphysics data enables PDE solvers to incorporate more comprehensive physical laws, leading to more accurate solutions to real-world problems.

## 1 Introduction

Partial Differential Equations (PDEs) are fundamental mathematical tools for modeling real-world physical systems and are integral to research across a wide range of disciplines, including electromagnetics, heat conduction, fluid dynamics, solid mechanics, acoustics, and mass transport. PDEs express constraint relationships among physical quantities in the spatiotemporal domain via differential operators, thereby establishing a bridge between fundamental physical principles and practical engineering applications (Evans, 2022). However, real-world physical systems frequently involve intricate interactions among multiple physical phenomena. Solving such PDEs typically requires the decoupling of the underlying physical processes, followed by the construction of a corresponding multiphysics model. These multiphysics PDEs are often highly nonlinear, with mathematical complexity arising from multiple equations, multiscale phenomena, numerous interacting variables, and strong bidirectional coupling—posing new challenges for efficient and accurate computation.

In recent years, Scientific Machine Learning (SciML) (Thiyagalingam et al., 2022) has demonstrated promising advances in PDE solving by leveraging data-driven methodologies and physical prior constraints, thereby extending the applicability of traditional numerical approaches such as finite element methods (FEM) (Reddy, 1993) and finite volume methods (FVM) (Eymard et al., 2000). Nonetheless, existing deep learning frameworks are confined to the isolated modeling of single physical domains—for instance, solving only the heat conduction equation to model thermal transfer (Edalatifar et al., 2021; Wang et al., 2021) or the elasticity equation for mechanical deformation (Abueidda et al., 2021). In fluid mechanics, although obtaining the velocity field requires simultaneous solution of both the continuity and momentum equations, this process does not inherently involve multiphysics coupling. (Miyanawala & Jaiman, 2017). Consequently, the efficacy of current architectures in multiphysics scenarios remains uncertain. Key open questions include whether these models can handle multiple interacting variables, whether they can effectively capture the strong nonlinear characteristics of inter-field coupling, and whether they are susceptible to error accumulation in the coupled terms. Although some foundational benchmark efforts have emerged in the context of machine learning-based PDE solvers, the vast majority are limited to single-physics problems. Currently, there exists no standardized benchmark tailored to multiphysics PDEs, posing substantial obstacles to systematic research and model evaluation. Constructing datasets for multiphysics problems is notably more challenging than for single-physics cases: it requires not only large-scale numerical simulations or experimental data but also the integration of domain-specific physical knowledge to formulate PDEs that accurately reflect true coupling mechanisms. Such demands pose particular challenges for researchers who may lack domain expertise in physics.

To address these challenges, this work introduces *Multiphysics Bench*, the first systematically designed benchmark framework tailored for machine learning-based solvers of multiphysics PDEs. This work makes three main contributions. **First,** we introduce the first general multiphysics benchmark dataset that encompasses six canonical coupled scenarios across domains such as electromagnetics, heat transfer, fluid flow, solid mechanics, pressure acoustics, and mass transport coupling. This benchmark features the most comprehensive set of coupling types, the largest diversity of PDE formulations, and the most extensive dataset scale currently available, thereby offering a valuable resource for accelerating model development and validation. **Second,** we conduct the first systematic evaluation and investigation on four representative machine learning approaches—Physics-Informed Neural Networks (PINNs) (Raissi et al., 2019b), Fourier Neural Operators (FNO) (Li et al., 2021), DeepONets (Lu et al., 2021), and DiffusionPDE (Huang et al., 2024) models —within multiphysics contexts. Our findings indicate that direct application of these single-physics methods to multiphysics problems yields suboptimal performance. **Third,** in our a comprehensive series of experiments and analyses, we report valuable insights into the application of machine learning techniques for solving multiphysics PDEs, such as a bag of tricks for handling multiphysics scenarios. By aligning computational solvers more closely with the governing physical laws, MultiphysicsBench provides a robust foundation for developing models that are more accurate, resilient, and generalizable—paving the way for future advances in simulating complex, coupled physical systems.

## 2 RELATED WORK

In this section, we review related work and discuss the position of our work.

**PDE Benchmark** Constructing a benchmark for multiphysics problems is a nontrivial and challenging endeavor. In contrast to traditional benchmarks based on images (Deng et al., 2009) or videos (Perazzi et al., 2016)—which are often universally applicable across a variety of tasks—datasets in computational physics are typically tailored to specific classes of PDEs. For single-physics PDE problems, several benchmark datasets have been developed. For instance, (Takamoto et al., 2022) investigates 11 fluid dynamics-related PDEs, focusing on both steady-state and transient velocity fields. Similarly, (Burark et al., 2024) presents a dataset comprising 10 continuous dynamical systems spanning fluid and solid mechanics, incorporating physical quantities such as velocity, stress, and displacement in both steady-state and transient regimes. In (Zheng et al., 2025), the authors construct five types of PDEs associated with inverse imaging, involving domains such as electromagnetism, optics, and fluid dynamics. The study includes steady-state electromagnetic fields, velocity fields, and mechanical wave fields. These efforts, however, are restricted to single-physics scenarios and do not consider the coupling between different physical fields. Although The Well (Ohana et al., 2024) assembles a large collection of physics datasets, it does not explicitly focus on multiphysics

problems and includes only a small fraction of multiphysics data. While a few open-source multi-physics datasets exist (Hassan et al., 2023), they typically contain a limited set of physical quantities. Consequently, the development of a benchmark dataset encompassing diverse coupled multiphysics scenarios is both necessary and valuable.

**Scientific Machine Learning for Solving PDEs** With the recent rapid advancement of SciML, increasing attention has been directed toward leveraging neural networks to approximate solutions of PDEs. Depending on the learning paradigm, current approaches can be categorized into four main types: traditional purely data-driven models, PINNs, operator learning frameworks, and generative architecture. Traditional models do not explicitly incorporate the governing PDEs; instead, they rely solely on data to train predictive models of physical dynamics (Raissi et al., 2019a). PINNs (Karniadakis et al., 2021) introduced by Raissi et al. (Raissi et al., 2017; Raissi et al., 2017), integrate PDE constraints into the loss function to guide the learning process automatic differentiation (Raissi et al., 2019b), finite difference schemes (Ren et al., 2022) or weak formulations (integral loss) (Eshaghi et al., 2025; Gao et al., 2022). Unlike pointwise regression models, operator learning approaches directly learn mappings between function spaces. Notable architectures include Deep-ONet (Lu et al., 2021), FNO (Li et al., 2021), Graph Neural Operator (GNO) (Anandkumar et al., 2019) and Poseidon (Herde et al., 2024). For example, DeepONet employs a dual-network structure composed of a Branch Net and a Trunk Net, while FNO utilizes the Fourier transform to efficiently model operators in the frequency domain. Generative models, on the other hand, aim to learn the mapping from a latent space to the PDE solution space. Canonical architectures include Generative Adversarial Networks (GANs) (Yang et al., 2020), Variational Autoencoders (VAEs) (Shin & Choi, 2023), and diffusion models (Huang et al., 2024). For instance, Diffusion-PDE models can generate plausible solutions in data-scarce regimes by learning the joint distribution of system parameters and physical fields. Nevertheless, most of these network architectures have been developed and validated primarily for single-physics problems. Their performance and suitability in multiphysics scenarios remain largely unexplored. This research gap highlights the need for a comprehensive benchmark to evaluate and compare the effectiveness of various SciML architectures in modeling coupled multiphysics systems.

## 3 MULTIPHYSICS BENCH

In this section, we formally introduce the Multiphysics Bench. We also discuss the six representative multiphysics problems and how they matter to real-world physical systems.

**Preliminary and Formulation** A coupled multiphysics system arises from the interaction and mutual influence of $M$ distinct physical fields, each governed by a partial differential equation (PDE). Such a system can be represented as a set of $M$ coupled PDEs. Let the solution domain be denoted by $\mathcal{D} = \mathcal{D}(\mathcal{X}, \mathcal{T})$, where the spatial variables satisfy $\mathbf{x} \in \mathcal{X} \subset \mathbb{R}^N$, the temporal variable satisfies $t \in \mathcal{T} \subset \mathbb{R}$, and the $i$th physical field is defined as $\mathbf{u}^i \in \mathcal{U} \subset \mathbb{C}^{N+1}$, for $i = 1, 2, \ldots, M$. The governing coupled PDE system can then be written as:

$$\mathcal{L}^i \left( \mathbf{u}^1, \mathbf{u}^2, \ldots, \mathbf{u}^M \right) (\mathbf{x}, t; \theta^i) = f^i(\mathbf{x}, t), \quad i = 1, 2, \ldots, M. \tag{1}$$

Here, $\mathcal{L}^i$ denotes the differential operator associated with the $i$-th equation, $\theta^i$ represents the corresponding parameter set, and $f^i$ denotes the external source term. To ensure a well-posed problem, appropriate boundary and initial conditions must be imposed as:

$$\mathcal{B}^i \left( \mathbf{u}^1, \mathbf{u}^2, \ldots, \mathbf{u}^M \right) (\mathbf{x} \in \partial\mathcal{X}, t; \theta^i) = B^i(\mathbf{x} \in \partial\mathcal{X}, t), \quad i = 1, 2, \ldots, M; \tag{2}$$

$$\mathcal{I}^i \left( \mathbf{u}^1, \mathbf{u}^2, \ldots, \mathbf{u}^M \right) (\mathbf{x}, t = t_0; \theta^i) = I^i(\mathbf{x}, t = t_0), \quad i = 1, 2, \ldots, M. \tag{3}$$

In our formulation, the boundary conditions $\mathcal{B}$ and initial conditions $\mathcal{I}$ are fixed. Consequently, solving the coupled PDE system is equivalent to constructing a mapping $\mathcal{F}$ from the known parameters $\theta$ and source terms $f$ to the corresponding physical fields $\mathbf{u}$. The motivation for employing deep learning frameworks in solving multiphysics problems lies in the ability of neural networks $\varphi$ to approximate the mapping $\mathcal{F}$ by learning a function $\mathcal{F}_\varphi$ that minimizes the discrepancy between predicted and reference solutions across training samples. Assuming $L$ is the loss function, the learning objective can be formalized as:

$$\hat{\varphi} = \arg\min_\varphi \sum_{i=1}^M L \left[ \mathcal{F}_\varphi(\theta, f) - \mathbf{u}^i \right]. \tag{4}$$

**Multiphysics Problems** Multiphysics problems typically involve bidirectional coupling among multiple physical processes such as electromagnetics, fluid dynamics, heat transfer, and solid mechanics. The system behavior is jointly determined by a set of interdependent PDEs. Compared with single-physics problems, multiphysics systems exhibit stronger nonlinearity, higher dimensionality, and cross-scale temporal and spatial characteristics. Coupling mechanisms among physical fields (e.g., thermo-mechanical, fluid–structure, and electro-thermal coupling) can further amplify system instability and uncertainty. From both numerical solutions and SciML perspectives, multiphysics problems present significantly greater challenges. Firstly, different physical fields often involve heterogeneous PDE types (elliptic, parabolic, and hyperbolic) and distinct characteristic scales, and their coupling requires traditional solvers to simultaneously satisfy the stability conditions of multiple physical processes. Secondly, continuity constraints at coupling interfaces impose additional constraints requirements, necessitating specialized discretization schemes, solver designs, and parallel computing strategies. Finally, the feedback effects across physical fields intensify error propagation, making multiphysics problems more sensitive to numerical stability and discretization accuracy. Therefore, multiphysics simulation is not merely a simple superposition of single-physics PDEs, but a system-level modeling problem with complex coupling structures and global consistency constraints, which imposes significantly higher demands on the generalization ability, convergence robustness, and cross-scale expression capability of algorithms.

In this study, we present the first meticulously designed benchmark for multiphysics problems. As exhibited in Figure 1, the benchmark encompasses six distinct and representative physical coupling scenarios.

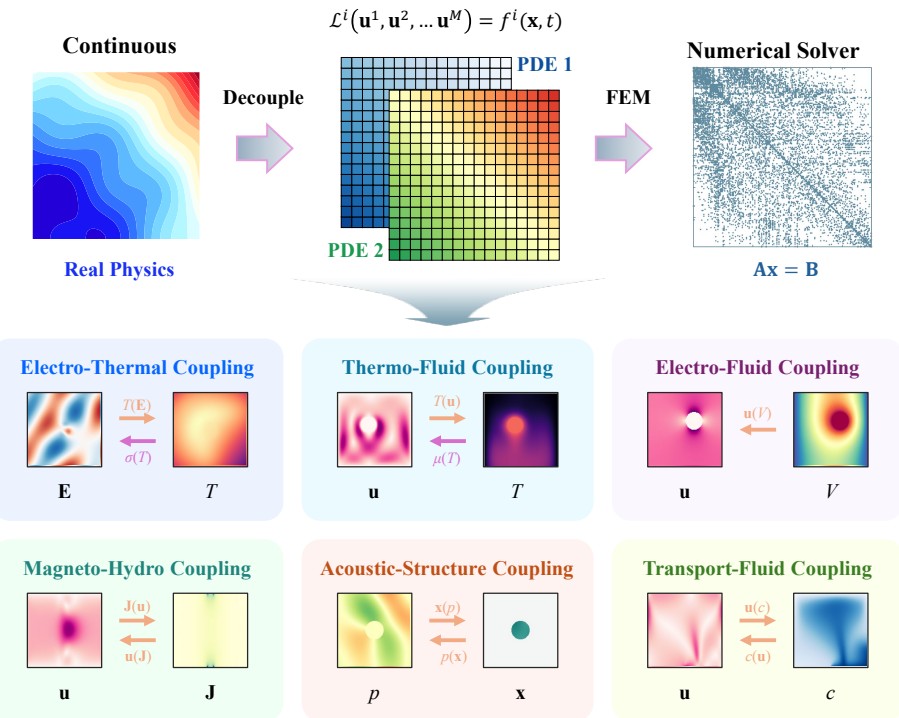

Figure 1: Illustration of the dataset collection process for the six multiphysics problems in the Multiphysics Bench. Each problem is abstracted from real-world physical phenomena, where the underlying PDEs and their coupling mechanisms are extracted and solved using FEM. Each subproblem involves the interaction between two coupled physical fields. Golden arrows indicate equation-level coupling, while pink arrows denote coupling via constitutive parameters.

Table 1: Summary of Multiphysics Bench's datasets with their respective Coupling model, Physics fields, Steady/transient state, Coupling type, and the numbers of Training/Testing samples.

| Model | Fields | State | Coupling Type | Training/Testing |
|---|---|---|---|---|
| Electro-Thermal | $\mathbf{E}, T$ | Frequency/Steady | Bidirectional | $10^4/10^3$ |
| Thermo-Fluid | $T, \mathbf{u}$ | Steady | Bidirectional | $10^4/10^3$ |
| Electro-Fluid | $\mathbf{u}, V$ | Steady | Unidirectional | $10^4/10^3$ |
| Magneto-Hydrodynamic | $\mathbf{J}, \mathbf{u}$ | Steady | Bidirectional | $10^4/10^3$ |
| Acoustic–Structure | $p, \mathbf{x}$ | Frequency | Bidirectional | $10^4/10^3$ |
| Mass Transport–Fluid | $c, \mathbf{u}$ | Transient | Bidirectional | $10^4/10^3$ |

**Overview** The proposed benchmark comprises five steady-state (frequency-domain) problems and one transient problems. For each dataset, samples are generated based on variations in geometric parameters (e.g., the position, major and minor axes, and orientation of elliptical objects) and physical source distributions (e.g., the center coordinates, amplitude, and standard deviation of Gaussian functions). Each coupled system involves between four and eight stochastic parameters, enabling the generation of diverse physical field distributions through parameter variation. The multiphysics coupling is implemented via two mechanisms: direct coupling of physical quantities within the governing equations, and indirect coupling through constitutive parameters and physical fields. The benchmark includes five bidirectionally coupled models and one unidirectionally coupled model. Furthermore, the benchmark incorporates a range of boundary conditions tailored to each physical system, such as Dirichlet and insulation conditions for electric potential fields, adiabatic and convective heat transfer conditions for temperature fields, slip-free and zero-flux conditions for velocity fields, and scattering boundary conditions for both acoustic pressure and electric fields. By integrating diverse data samples, equation types, coupling strategies, and boundary conditions, the benchmark significantly enhances the quality and utility of the dataset, thereby facilitating rigorous evaluation of future multiphysics PDE models. A brief overview of the six multiphysics scenarios and their associated PDEs is provided in Table 1, with further details available in Appendix A.1.

**Electro-Thermal Coupling** (Wave Equation and Heat Conduction Equation): Electro-thermal coupling refers to the interaction between the electric field $\mathbf{E}$ and the temperature field $T$. When electromagnetic waves incident upon a conductive structure, they induce currents within the material. These currents generate Joule heating, which serves as an internal heat source and consequently elevates the temperature. This phenomenon is described by the following equations:

$$\nabla^2\mathbf{E} + k_0^2\mu_r\varepsilon_r\left(1 + \frac{\sigma}{j\omega\varepsilon_0}\right)\mathbf{E} = \mathbf{0}, \ \nabla\cdot(\kappa\nabla T) + \frac{1}{2}\sigma\left|\mathbf{E}\right|^2 = 0. \tag{5}$$

Here, $\mu_r$, $\varepsilon_r$, and $\varepsilon_0$ denote the relative permeability, relative permittivity, and vacuum permittivity, respectively. $k_0$ is the wavenumber in vacuum, $\omega$ is the angular frequency, and $\kappa$ and $\sigma$ represent thermal and electrical conductivity, respectively. This coupling mechanism underpins various applications, including thermal management in electronic components (Zaman et al., 2021; Yang et al., 2024) (e.g., microprocessors), inductive heating (Istardi & Triwinarko, 2011) (e.g., welding and metal processing), biomedical technologies (Adam & Hashim, 2014) (e.g., photothermal therapy), and aerospace engineering (Wang et al., 2022b; Nenarokomov et al., 2017) (e.g., radiative heating of satellite surfaces).

**Thermo-Fluid Coupling** (Navier–Stokes Equations and Heat Balance Equation): This model characterizes the steady-state natural convection within enclosed cavities, emphasizing the coupling between the temperature field $T$ and the velocity field $\mathbf{u}$. A localized heat source elevates the temperature of the surrounding air, generating density gradients that, in turn, induce buoyancy forces and drive fluid motion (Wang et al., 2022a; Wang, 2024). The governing equations are as follows:

$$\rho\left(\mathbf{u}\cdot\nabla\right)\mathbf{u} = -\nabla p + \mu\nabla^2\mathbf{u} + \rho\mathbf{g}, \ \nabla\cdot(\rho\mathbf{u}) = 0, \ \rho C_p\mathbf{u}\cdot\nabla T - \nabla\cdot(\kappa\nabla T) = Q. \tag{6}$$

In these expressions, $p$ is the pressure, $\mu$ is the dynamic viscosity, $\rho$ is the fluid density, $C_p$ is the specific heat capacity at constant pressure, $\kappa$ denotes thermal conductivity, and $Q$ represents the volumetric heat source. Thermo-Fluid coupling is essential in the design and optimization of systems such as electronic cooling (Bai, 2024) (e.g., chip heat dissipation), energy systems (Mylonakis et al.,

2014) (e.g., nuclear reactor cooling), and precision thermal control in manufacturing (Michopoulos et al., 2018).

**Electro-Fluid Coupling** (Navier–Stokes Equations and Current Continuity Equation): This model describes electroosmotic flow within porous media, involving the interaction between the fluid velocity field $\mathbf{u}$ and the electric potential field $V$. An applied electric potential generates an electric field, which exerts a Coulomb force on the fluid, resulting in flow. The system is governed by:

$$\nabla \cdot \mathbf{u} = 0, \ \nabla \cdot (\sigma \nabla V) = 0, \ \mathbf{u} = -\frac{\epsilon_p a^2}{8\mu}\nabla p + \frac{\epsilon_p \epsilon_w \zeta}{\mu}\nabla V. \tag{7}$$

In the third equation of Equation 7, the first and second terms correspond to pressure-driven and electroosmotic components of fluid motion, respectively. Here, $\epsilon_p$ denotes porosity, $a$ is the average pore radius, $\mu$ is the dynamic viscosity, $\epsilon_w$ is the dielectric permittivity, $\zeta$ is the zeta potential, and $p$ and $V$ represent pressure and electric potential. This coupling is foundational to applications such as micropumps and micromixers in microfluidic systems (Annabestani et al., 2020).

**Magneto-Hydrodynamic Coupling** (Ampère's Law, Continuity Equation, Navier–Stokes Equations, and Lorentz Force): This model addresses the interaction between electromagnetic fields and fluid dynamics, characterized by coupling between the current density field $\mathbf{J}$ and velocity field $\mathbf{u}$. When conductive fluids are subjected to electric and magnetic fields, the resulting currents experience magnetic deflection, influencing the fluid motion. The system is described by:

$$\mathbf{J} = \sigma\left(-\nabla V + \mathbf{u} \times \mathbf{B}\right), \ \nabla \cdot \mathbf{J} = 0, \ \nabla \times \mathbf{B} = \mu_s \mathbf{J}, \ \rho\left(\mathbf{u} \cdot \nabla\right)\mathbf{u} = -\nabla p + \mu\nabla^2\mathbf{u} + \mathbf{J} \times \mathbf{B}. \tag{8}$$

Here, $\sigma$ is electrical conductivity, $V$ is electric potential, $\mathbf{B}$ is the magnetic flux density, $\mu_s$ is magnetic permeability, and $\rho$ and $\mu$ denote fluid density and dynamic viscosity, respectively. The term $\mathbf{J} \times \mathbf{B}$ represents the Lorentz force. This model finds extensive application in electromagnetic pumps, plasma confinement devices (e.g., tokamaks) (Štancar et al., 2019), astrophysical phenomena (Roman & Fireteanu, 2011), and pollutant transport modeling (Cristea et al., 2010).

**Acoustic–Structure Coupling** (Acoustic Wave and Structural Vibration Equation): This model examines the mutual interaction between acoustic waves and elastic solids, particularly the coupling among acoustic pressure $p$ and structural displacement $\mathbf{x}$. When an acoustic wave interacts with an elastic structure immersed in a fluid, it induces structural vibrations, which in turn radiate secondary acoustic waves. This bidirectional interaction is governed by the following coupling equations:

$$\nabla \cdot \left(\frac{1}{\rho_c}\nabla p\right) + \frac{\omega^2 p}{\rho_c c_c^2} = 0, \ -\rho\omega^2\mathbf{x} = \nabla \cdot \mathbf{S}, \ \mathbf{n} \cdot \left(\frac{1}{\rho_0}\nabla p\right) = \left(\mathbf{n} \cdot \mathbf{x}\right)\omega^2. \tag{9}$$

Here, $\omega$ is angular frequency, $\rho_c$ and $c_c$ represent fluid density and speed of sound, respectively. Besides, $\rho$ and $\mathbf{S}$ are the density and stress tensor of the solid, while $\mathbf{n}$ is the unit normal vector on the interface. The coupling condition on the interface ensures continuity of acoustic pressure and structural displacement. This model plays a fundamental role in the design and analysis of acoustic systems such as loudspeakers (Magalotti, 2015), medical ultrasound devices (Kyriakou, 2015), and acoustic sensors (Draper et al., 2023).

**Mass Transport–Fluid Coupling** (Darcy's Law and Convection–Diffusion Equation): This model captures the transient transport of dilute solutes in porous media, emphasizing coupling between velocity field $\mathbf{u}$ and concentration field $c$. Concentration variations produce density gradients that drive buoyancy-induced flow, thereby enhancing solute transport. The governing equations are:

$$\beta\frac{\partial \epsilon_p c}{\partial t} + \nabla \cdot \left[(\rho_0 + \beta c)\mathbf{u}\right] = 0, \ \frac{\partial(\theta_s c)}{\partial t} + \mathbf{u} \cdot \nabla c - \nabla \cdot (\theta_s \tau D_L \nabla c) = S_c. \tag{10}$$

Here, $\beta$ is the solutal expansion coefficient, $\epsilon_p$ is porosity, $\rho_0$ is the density of pure water, $\theta_s$ is the fluid volume fraction, $\tau$ is tortuosity, $D_L$ is the dispersion coefficient, and $S_c$ is the source term. This coupling framework is widely used in environmental engineering (Niessner & Helmig, 2009) (e.g., contaminant transport in groundwater), petroleum engineering (Han et al., 2024) (e.g., multi-phase diffusion) and biomedical applications (Moradi Kashkooli et al., 2024) (e.g., drug delivery in tissues).

**Dataset Collection**   All PDEs in this work are solved using the forward numerical solver—FEM. The overall data generation process is illustrated in Figure 1. Details of the numerical implementation and network-specific settings are provided in Appendix A.2.

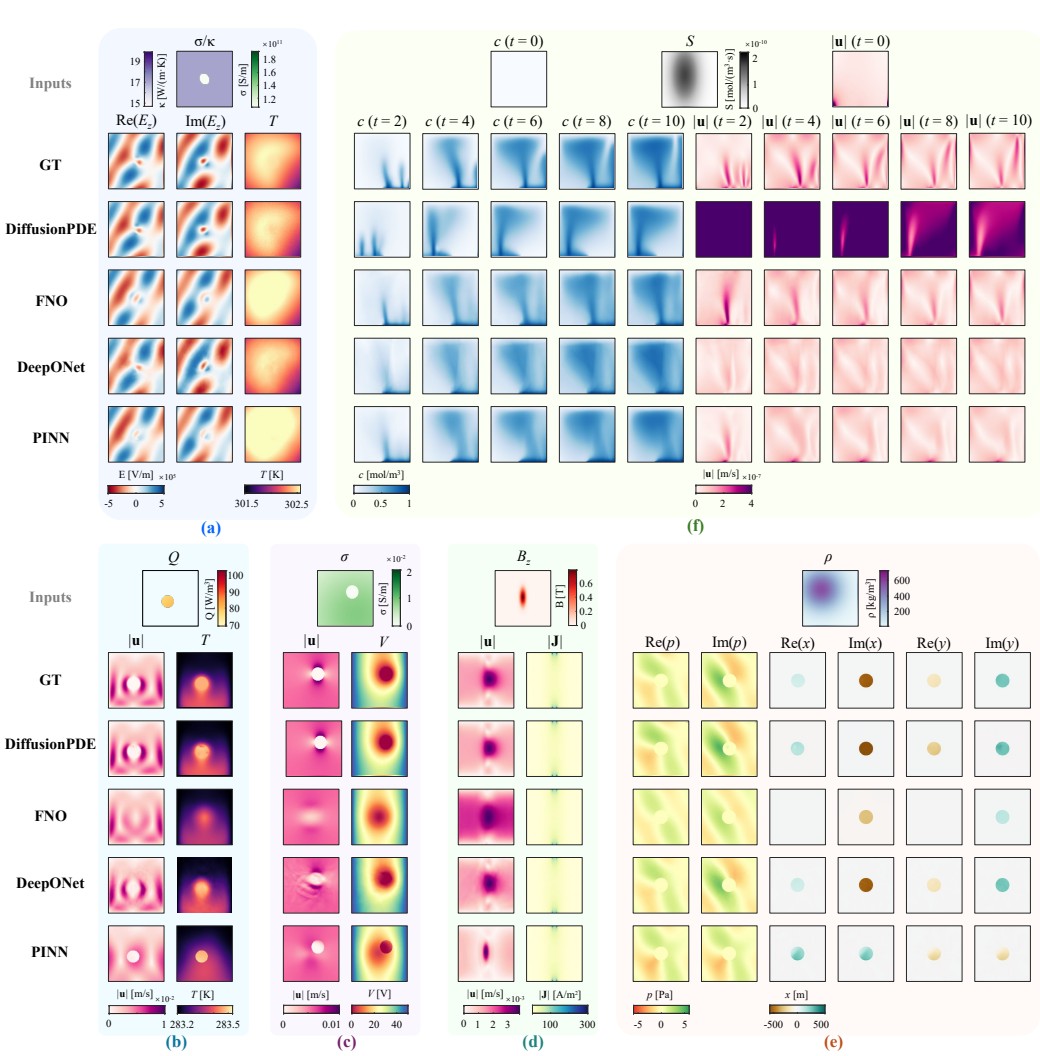

Figure 2: Performance comparisons of different frameworks on benchmark multiphysics problems. (a) Electro-Thermal Coupling, (b) Thermo-Fluid Coupling, (c) Electro-Fluid Coupling, (d) Magneto-Hydrodynamic (e) Acoustic–Structure Coupling, (f) Mass Transport–Fluid Coupling.

## 4 EMPIRICAL ANALYSIS

In this section, we empirically investigate representative SciML baselines on Multiphysics Bench.

**Problem Setting**   We conduct empirical studies on six representative multiphysics PDE problems using four diverse SciML architectures. Each dataset is generated with a fixed spatial resolution of $128 \times 128$. The number of training and testing samples, along with the architecture-specific implementation details for each problem, are summarized in Table 1 and elaborated in Appendix A.2.

**Baseline Setup**   We train and evaluate four representative machine learning-based PDE solvers on the benchmark: PINNs, DeepONet, FNO, and DiffusionPDE. To ensure a fair comparison, we selected hyperparameters to keep the model scale (i.e., number of trainable parameters) approximately consistent across baselines. Based on this setup, we conducted grid searches for key hyperparameters such as learning rate and training epochs to optimize performance. For PINNs, we incorporate residual blocks and impose physics-informed PDE loss, with a PDE-to-data loss weight ratio of 1:1000. DeepONet adopts an unstacked architecture with a basis dimension $p = 256$. FNO is implemented with 12 Fourier modes using its default configuration. DiffusionPDE leverages score matching to reconstruct the final state from initial conditions. For fair comparison in terms of efficiency, we use the Euler method with 10 inference steps during prediction. Comprehensive details on training configurations and hyperparameters are provided are provided in Appendix B.2. All source code and experimental configurations are released to ensure reproducibility and facilitate future research.

**Baseline Performance**   Figure 2 shows the performance of the four architectures across the six benchmark multiphysics tasks, with corresponding relative $L_2$ errors summarized in Table 2. Additional results for extended samples are shown in Figures 15–20 in Appendix C.1. Tables 14–19 provide detailed comparisons of average prediction errors under multiple evaluation metrics defined in B.1. To account for possible disturbances in real-world scenarios, noise was intentionally added. Table 20 provides an error analysis for the noisy case, using the Electro-Thermal coupling problem as an example. Our analysis shows that for the first five steady-state or frequency-domain problems, all four architectures achieve generally acceptable performance. Among them, DeepONet—which is grounded in the general operator approximation framework—consistently yields the most accurate predictions across most coupled fields. In contrast, DiffusionPDE produces statistically plausible field distributions but suffers from substantial pointwise errors, likely due to the stochastic noise intrinsic to the generative modeling process. FNO often fails in steady-state scenarios, frequently producing collapsed and non-diverse outputs—potentially due to challenges in optimizing in the frequency domain, especially when high spectral similarity exists across samples. PINNs often exhibit discontinuities near interface regions, which may stem from truncation errors introduced by finite-order differential operators. All models exhibit degraded performance in transient Mass Transport–Fluid coupling tasks. DiffusionPDE, in particular, fails to generate accurate spatial predictions, likely due to its limited capacity to adapt to the increasing complexity of systems with multiple unknown variables. Furthermore, the current network architectures appear insufficiently expressive for capturing time-dependent multiphysics dynamics. Computational cost analysis, including GPU memory consumption and runtime, is reported in Section C.3, where DiffusionPDE incurs the highest computational overhead. The details of the training for Multiphysics Bench and the adjustments to the baseline models, as well as their limitations, are presented in Appendix C.1.

**Complete vs. Incomplete Physical Priors**   Real-world physical phenomena are often governed by interdependent processes such as Electro-Thermal, Thermo-Fluid, or Acoustic–Structural interactions, which are naturally described using coupled PDEs across different physical domains. However, in practice, only partial knowledge of the governing physics may be available—for example, observations of a single field governed by a subset of the full PDE system. For instance, in an Electro-Thermal system, we might only observe the temperature field $T$ governed by the heat equation, while lacking access to the electric field $\mathbf{E}$. To quantify the impact of incomplete physical priors, we design a case study where the underlying system is jointly governed by the temperature field $T$ and the electric field $\mathbf{E}$, satisfying the coupled Equation (5). During training, however, we constrain supervision to the scalar field $T$, using only a simplified heat conduction equation:

$$\nabla \cdot (\kappa \nabla T) = 0. \tag{11}$$

Table 2: Baseline model performance on the six subtasks of Multiphysics Bench. Metric: Relative $L_2$ Error. $\bar{\cdot}$ denotes the average RE across the components of each physical field.

| Subtask | Fields | PINNs | DeepONet | FNO | DiffusionPDE |
|---------|--------|-------|----------|-----|--------------|
| Elec-Ther | $\bar{E}_z$ | $4.428 \times 10^{-1}$ | $\mathbf{2.595 \times 10^{-1}}$ | $4.431 \times 10^{-1}$ | $5.415 \times 10^{-1}$ |
| | $T$ | $1.573 \times 10^{-3}$ | $2.317 \times 10^{-3}$ | $\mathbf{1.482 \times 10^{-3}}$ | $1.560 \times 10^{-3}$ |
| Ther-Flu | $\bar{u}$ | $5.612 \times 10^{-1}$ | $\mathbf{1.022 \times 10^{-1}}$ | $4.353 \times 10^{-1}$ | $1.944 \times 10^{-1}$ |
| | $T$ | $1.330 \times 10^{-4}$ | $\mathbf{1.760 \times 10^{-5}}$ | $1.419 \times 10^{-4}$ | $1.060 \times 10^{-4}$ |
| Elec-Flu | $V$ | $1.831 \times 10^{-1}$ | $\mathbf{1.962 \times 10^{-2}}$ | $2.368 \times 10^{-1}$ | $3.520 \times 10^{-1}$ |
| | $\bar{u}$ | $4.705 \times 10^{-1}$ | $\mathbf{2.453 \times 10^{-1}}$ | $5.112 \times 10^{-1}$ | $8.347 \times 10^{-1}$ |
| MHD | $\bar{J}$ | $1.377 \times 10^{-1}$ | $4.346 \times 10^{-1}$ | $8.893 \times 10^{-2}$ | $\mathbf{8.655 \times 10^{-2}}$ |
| | $\bar{u}$ | $3.055 \times 10^0$ | $\mathbf{1.252 \times 10^{-1}}$ | $1.723 \times 10^0$ | $4.608 \times 10^0$ |
| Acous–Struc | $\mathrm{Re}(p)$ | $5.029 \times 10^{-1}$ | $\mathbf{6.184 \times 10^{-2}}$ | $4.965 \times 10^{-1}$ | $4.653 \times 10^{-1}$ |
| | $\mathrm{Re}(\bar{x})$ | $3.661 \times 10^0$ | $\mathbf{3.201 \times 10^{-1}}$ | $1.772 \times 10^0$ | $7.694 \times 10^0$ |
| MT-Flu | $\bar{u}$ | $\mathbf{1.716 \times 10^{-1}}$ | $6.811 \times 10^{-1}$ | $5.945 \times 10^{-1}$ | $4.528 \times 10^0$ |
| | $c$ | $2.713 \times 10^{-1}$ | $\mathbf{2.617 \times 10^{-1}}$ | $2.623 \times 10^{-1}$ | $8.340 \times 10^{-1}$ |

As shown in Table 3, all models demonstrate degraded performance under this single-field supervision, despite the data reflecting full multiphysics interactions. A complementary experiment predicting $E_z$ without access to $T$ (see Appendix C.2) yields similar outcomes. These findings underscore the importance of incorporating complete multiphysics knowledge during training: models trained on full field observations are better able to capture inter-domain structures and generalize across coupled fields, either implicitly or through explicit PDE constraints.

Table 3: Multiphysics data incorporates more comprehensive physical priors and demonstrates superior performance. Metric: Relative $L_2$ Error. Dataset: Electro-thermal.

| | PINNs | DeepONet | FNO | DiffusionPDE |
|---|-------|----------|-----|--------------|
| $T$ (Incomplete) | $1.59 \times 10^{-3}$ | $\mathbf{1.59 \times 10^{-3}}$ | $2.02 \times 10^{-2}$ | $2.04 \times 10^{-3}$ |
| $T$ (Complete) | $\mathbf{1.57 \times 10^{-3}}$ | $2.32 \times 10^{-3}$ | $\mathbf{1.48 \times 10^{-3}}$ | $\mathbf{1.56 \times 10^{-3}}$ |

**Data Scaling**   To assess data scalability, we evaluate the four baselines on the Electro-Thermal task across varying training sizes (300 to 30,000 samples). As reported in Table 4, prediction accuracy does not consistently and significantly improve with increasing data volume, revealing the absence of smooth scaling behavior. The scaling results of Thermo-Fluid are presented in Appendix C.4. Note that data scaling law are considered as an important advantage of foundations models. Moreover, performance varies considerably across different physical fields, suggesting limited model adaptability to complex multiphysics interactions. These results highlight two key challenges in current SciML pipelines: (a) **Low data efficiency**, where SciML models performance often get saturated around 10,000 samples. (b) **Limited generalization across physical scales**, underscoring the need for more scalable and physically grounded foundation models capable of leveraging larger datasets and capturing cross-domain dynamics in a scale-invariant manner.

## 5   DISCUSSION AND CONCLUSION

**Tricks for Solving Multiphysics PDEs**   While current SciML solvers perform well on single-physics tasks, their effectiveness deteriorates in multiphysics settings due to strong inter-field coupling. To mitigate these challenges, we identify key bottlenecks and propose lightweight yet effective tricks to enhance model robustness and accuracy. Details are presented in Appendix C.5.

**Limitations and Analysis of Existing Models**   Our systematic evaluation of four deep learning frameworks for multiphysics modeling identifies several key limitations: (1) Existing architectures

Table 4: Relative $L_2$ error performance of baseline models across varying data scales. Metric: Relative $L_2$ Error. Dataset: Electro-thermal. D-ONet: DeepONet, Diff-PDE: DiffusionPDE.

| RE | Fields | 300 | 1,000 | 3,000 | 10,000 | 30,000 |
|---|---|---|---|---|---|---|
| PINNs | $\mathrm{Re}(E_z)$ | $6.75 \times 10^{-1}$ | $4.71 \times 10^{-1}$ | $4.51 \times 10^{-1}$ | $4.44 \times 10^{-1}$ | $\mathbf{4.42 \times 10^{-1}}$ |
| | $\mathrm{Im}(E_z)$ | $7.26 \times 10^{-1}$ | $4.92 \times 10^{-1}$ | $4.64 \times 10^{-1}$ | $\mathbf{4.47 \times 10^{-1}}$ | $4.48 \times 10^{-1}$ |
| | $T$ | $1.55 \times 10^{-3}$ | $1.58 \times 10^{-3}$ | $1.60 \times 10^{-3}$ | $1.57 \times 10^{-3}$ | $\mathbf{1.52 \times 10^{-3}}$ |
| D-ONet | $\mathrm{Re}(E_z)$ | $4.66 \times 10^{-1}$ | $4.56 \times 10^{-1}$ | $3.94 \times 10^{-1}$ | $2.57 \times 10^{-1}$ | $\mathbf{2.53 \times 10^{-1}}$ |
| | $\mathrm{Im}(E_z)$ | $4.90 \times 10^{-1}$ | $4.65 \times 10^{-1}$ | $3.96 \times 10^{-1}$ | $2.62 \times 10^{-1}$ | $\mathbf{2.56 \times 10^{-1}}$ |
| | $T$ | $1.81 \times 10^{-3}$ | $\mathbf{1.57 \times 10^{-3}}$ | $2.29 \times 10^{-3}$ | $2.32 \times 10^{-3}$ | $2.26 \times 10^{-3}$ |
| FNO | $\mathrm{Re}(E_z)$ | $6.51 \times 10^{-1}$ | $4.51 \times 10^{-1}$ | $4.46 \times 10^{-1}$ | $4.45 \times 10^{-1}$ | $\mathbf{4.45 \times 10^{-1}}$ |
| | $\mathrm{Im}(E_z)$ | $8.34 \times 10^{-1}$ | $4.58 \times 10^{-1}$ | $4.56 \times 10^{-1}$ | $4.55 \times 10^{-1}$ | $\mathbf{4.54 \times 10^{-1}}$ |
| | $T$ | $1.48 \times 10^{-3}$ | $1.47 \times 10^{-3}$ | $1.47 \times 10^{-3}$ | $1.47 \times 10^{-3}$ | $\mathbf{1.46 \times 10^{-3}}$ |
| Diff-PDE | $\mathrm{Re}(E_z)$ | $6.41 \times 10^{-1}$ | $6.28 \times 10^{-1}$ | $6.12 \times 10^{-1}$ | $5.61 \times 10^{-1}$ | $\mathbf{5.28 \times 10^{-1}}$ |
| | $\mathrm{Im}(E_z)$ | $6.54 \times 10^{-1}$ | $5.95 \times 10^{-1}$ | $5.99 \times 10^{-1}$ | $\mathbf{5.41 \times 10^{-1}}$ | $5.62 \times 10^{-1}$ |
| | $T$ | $1.63 \times 10^{-3}$ | $1.75 \times 10^{-3}$ | $1.93 \times 10^{-3}$ | $1.67 \times 10^{-3}$ | $\mathbf{1.63 \times 10^{-3}}$ |

inadequately represent cross-field interactions, restricting their ability to capture bidirectional and nonlinear couplings intrinsic to multiphysics systems. (2) High-frequency features are poorly preserved; FNO and related frequency-domain methods underrepresent fine-scale variations, while PDE loss-based approaches suffer from resolution constraints, diminishing accuracy in regions with steep gradients. (3) Gradient inconsistencies arise in multiphysics training, as models like PINNs and DiffusionPDE generate conflicting gradient directions and magnitudes, leading to instability, field dominance shifts, and potential negative transfer. (4) Inference accuracy deteriorates with increased diffusion steps in DiffusionPDE, where heavy reliance on guidance signals exacerbates error accumulation and noise amplification, compromising prediction quality and reliability.

**Conclusion** While SciML for solving PDEs has attracted great attention recently, previous SciML studies overlooked important multiphysics problems, as most real-world physical systems actually involve in coupled multiple physical phenomena. In this work, we proposed ***Multiphysics Bench***, the most comprehensive multiphysics PDE dataset to date, featuring the broadest range of coupling types, the greatest diversity of multiphysics PDE formulations, and the largest scale of coupled physics data. With the proposed benchmark, we conducted the first systematic investigation on representative SciML baselines for solving multiphysics PDEs, producing multiple useful tricks that improve the baseline performance in multiphysics scenarios. Our work establishes the first foundation in this direction and provides an insightful outlook for future research. We believe our work will motivate more physicists, mathematicians, and AI researchers to promote the development and applications of SciML for multiphysics.

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

## A  DATASET COLLECTION DETAILS

### A.1  MULTIPHYSICS PROBLEM DETAILS

Detailed configurations for solving multiphysics PDE—including materials, boundary conditions, and other settings—are provided here.

**Electro-thermal Coupling**

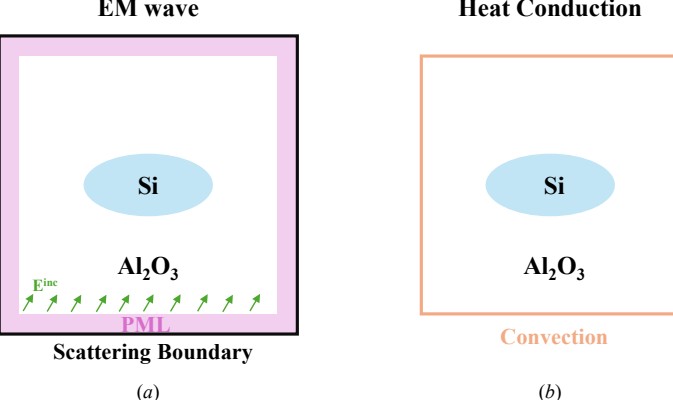

Figure 3: Simulation model for electro-thermal coupling. (a) Electromagnetic field model; (b) Thermal conduction model.

The electro-thermal coupling simulation model is presented in Figure 3, where panel (a) illustrates the electromagnetic model, and panel (b) depicts the heat conduction model. An elliptical silicon semiconductor pillar, characterized by a semi-major axis $a$, semi-minor axis $b$, and rotation angle $\varphi$, is embedded within a $128 \times 128$ mm square alumina substrate. The temperature-dependent electrical conductivity of silicon, $\sigma(T)$, is defined as follows:

$$\sigma\left(T\right) = \mu n q e^{-\frac{E_g}{k_B T}} = \sigma_0 e^{-\frac{E_g}{k_B T}}, \tag{12}$$

where $\mu$, $n$, and $q$ denote the carrier mobility, carrier concentration, and elementary charge, respectively. The energy band gap of silicon, $E_g$, is taken as 1.12 eV, while $k_B$ represents the Boltzmann constant. The exterior of the electromagnetic solution domain is enveloped by a 10 mm-thick perfectly matched layer (PML), designed to absorb scattered electromagnetic waves. The outermost boundary adheres to the scattering boundary condition:

$$\nabla \times (\nabla \times \mathbf{E}) - jk\mathbf{n} \times (\mathbf{E} \times \mathbf{n}) = 0, \tag{13}$$

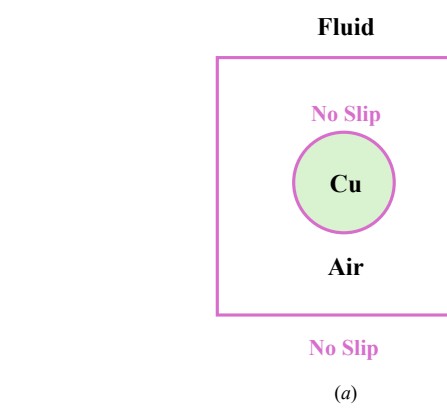 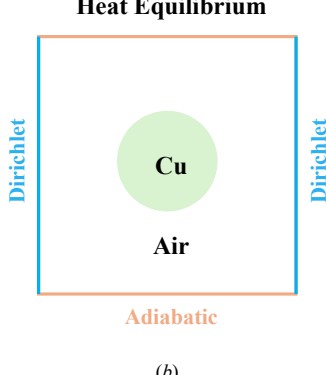

Figure 4: Simulation model for Thermo-Fluid coupling. (a) Fluid flow model; (b) Heat transfer model.

where $k$ denotes the wave number and $\mathbf{n}$ is the outward normal vector of the boundary. The incident electromagnetic wave is TE$_z$-polarized, with the electric field expressed as $E = \mathbf{z}E_m e^{jk(x\cos\theta + y\sin\theta)}$. In the thermal conduction model, the outer boundary is assigned a convective heat transfer condition:

$$-\kappa\frac{\partial T}{\partial \mathbf{n}} = h\left(T_{ext} - T\right),\tag{14}$$

where $\kappa$ represents the thermal conductivity, $h$ is the heat transfer coefficient, and $T_{ext}$ is the ambient temperature. As the electrical conductivity exhibits temperature dependence and the heat source is influenced by the electric field, a bidirectional coupling exists between the electric field and temperature. Upon constructing the physical model, a dataset comprising 11,000 samples was generated using the finite element method. The corresponding parameter settings are outlined in Table 5. The computed real and imaginary components of the $z$-direction electric field, along with the temperature field, are discretized into three distinct $128 \times 128$ matrices.

Table 5: Parameters of the Electro-Thermal Coupling Model

| Symbol | Physical Quantity | Value / Range |
|---|---|---|
| $a$ | Semi-major axis | [20, 30] m |
| $b$ | Semi-minor axis | [10, 20] m |
| $\varphi$ | Ellipse rotation angle | [0, 360°] |
| $\sigma_0(\text{Si})$ | Electrical conductivity of silicon at room temperature | [1.602, 4.806] $\times 10^{11}$ S |
| $\sigma_0(\text{Al}_2\text{O}_3)$ | Electrical conductivity of alumina | $1 \times 10^{-7}$ S |
| $k$ | Vacuum wave number | 83.73 rad/m |
| $E_m$ | Electric field amplitude | $3 \times 10^5$ V/m |
| $\theta$ | Incident field angle | $\pi/3$ rad |
| $\kappa(\text{Si})$ | Thermal conductivity of silicon | 70 W/(m·K) |
| $\kappa(\text{Al}_2\text{O}_3)$ | Thermal conductivity of alumina | [10, 20] W/(m·K) |
| $\varepsilon_r(\text{Si})$ | Relative permittivity of silicon | 11.7 |
| $\varepsilon_r(\text{Al}_2\text{O}_3)$ | Relative permittivity of alumina | 1 |
| $T_{\text{ext}}$ | Ambient temperature | 293.15 K |
| $h$ | Heat transfer coefficient | 15 W/(m²·K) |

**Thermo-Fluid Coupling** The basic model of Thermo-Fluid coupling is illustrated in Figure 4, where (a) and (b) represent the fluid model and the thermal equilibrium model, respectively. A circular copper pillar with radius $r$ and center coordinates $x_c, y_c$ is placed inside a square container with side length $L = 12.8$ cm. The container is filled with air, whose thermal conductivity $\kappa$ and

specific heat capacity $C_\rho$ are defined as:

$$\kappa = \sum_{i=0}^{4} \kappa_i T^i, \tag{15}$$

$$C_\rho = \sum_{i=0}^{4} C_i T^i, \tag{16}$$

$$\mu = \sum_{i=0}^{4} \mu_i T^i, \tag{17}$$

where $\kappa_i$, $C_i$, and $\mu_i$ are temperature-independent coefficients. For the fluid model, the no-slip boundary condition is applied on both the container walls and the surface of the copper pillar:

$$\mathbf{u} = 0. \tag{18}$$

For the thermal balance model, the left and right walls of the container are specified with Dirichlet boundary conditions, while the top and bottom walls are thermally insulated:

$$T = T_c, \ ( \ x = 0 \ \& \ x = L \ ), \tag{19}$$

$$\frac{\partial T}{\partial \mathbf{n}} = 0, \ ( \ y = 0 \ \& \ y = L \ ), \tag{20}$$

where $\mathbf{n}$ denotes the unit outward normal vector. The volumetric heat source inside the copper pillar is defined as:

$$Q = Q_0 e^{x+y}. \tag{21}$$

According to Equation 6, the temperature field $T$ and the velocity field $\mathbf{u}$ are bidirectionally coupled. After constructing the physical model, we generated a dataset with 11,000 samples using the finite element method. The corresponding parameter settings are listed in Table 6. The computed velocity components in the $x$-direction and $y$-direction, along with the temperature field, are discretized into three separate $128 \times 128$ matrices.

Table 6: Parameters of the Thermo-Fluid Coupling Model

| Symbol | Physical Quantity | Value / Range |
|---|---|---|
| $L$ | Side length of the container | 12.8 cm |
| $r$ | Radius of copper pillar | [1.28, 1.6] cm |
| $x_c, y_c$ | Center coordinates of the pillar | [-1.6, 1.6] cm |
| $\kappa_{\mathrm{Cu}}$ | Thermal conductivity of copper | 400 W/(m·K) |
| $\rho$ | Air density | 1.24 kg/m$^3$ |
| $Q_0$ | Heat source coefficient | 70 W/m$^3$ |
| $\mathbf{g}$ | Gravitational acceleration | 9.8 m/s$^2$ |

**Electro-Fluid Coupling**

The basic model of Electro-Fluid coupling is illustrated in Figure 5. The computational domain is a square region with side length $L = 1$ m, within which an elliptical region—with semi-major axis $a_{\mathrm{ellipse}}$ and semi-minor axis $b_{\mathrm{ellipse}}$ is removed. The entire domain is immersed in a conductive fluid, whose electrical conductivity $\sigma$ is defined as a Gaussian function of spatial position:

$$\sigma = \sigma_0 e^{-\frac{(x-x_0)^2 + (y-y_0)^2}{2s^2}}. \tag{22}$$

Here, $\sigma_0$, $x_0/y_0$ and $s$ represent the amplitude, center location, and spatial dispersion of the conductivity distribution, respectively. For the electrical boundary conditions, Dirichlet conditions are applied on the internal elliptical boundary and the left/right boundary of the domain:

$$V = V_0, \ x = \pm L/2, \tag{23}$$

$$V = 0, \ \text{Internal Ellipse}, \tag{24}$$

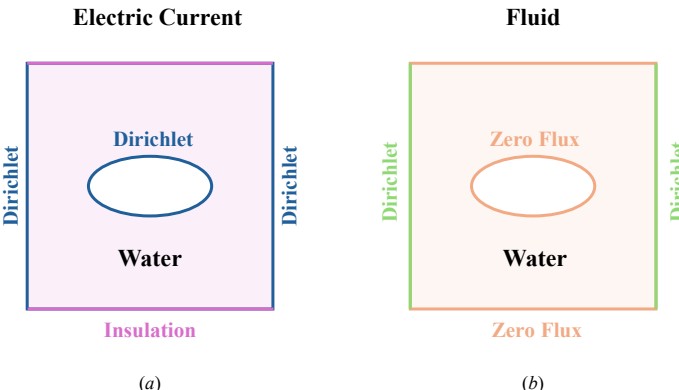

Figure 5: Simulation model for Electro-Fluid coupling. (a) Electrical model; (b) Fluid dynamics model.

$$\mathbf{n} \cdot \mathbf{J} = 0, \; x = \pm L/2. \tag{25}$$

Here, $\mathbf{n}$ denotes the unit outward normal vector. For the fluid boundary conditions, Dirichlet conditions are applied on the left and right boundaries for pressure, while zero-flux Neumann conditions are imposed on the top and bottom boundaries:

$$p = 0, \; x = -L/2, \tag{26}$$

$$p = p_0, \; x = L/2, \tag{27}$$

$$\frac{\partial p}{\partial \mathbf{n}} = 0, \; y = \pm L/2. \tag{28}$$

Since the velocity field depends on the electric potential while the potential is independent of the velocity, the coupling between them is unidirectional. After constructing the two physical models, we generated a total of 11,000 datasets using the finite element method. The corresponding parameter settings are provided in Table 7. The computed $x$- and $y$-components of the velocity field, together with the electric potential, are discretized into three separate $128 \times 128$ matrices, which serve as the learning targets for the neural network.

Table 7: Parameters of the Electrical-Fluid Coupling Model

| Symbol | Physical Quantity | Value / Range |
|---|---|---|
| $L$ | Side length of the container | 1.28 mm |
| $a_{\text{ellipse}}$ | Semi-major axis of ellipse | $[L/10, L/8]$ |
| $b_{\text{ellipse}}$ | Semi-minor axis of ellipse | $[L/10, L/8]$ |
| $x_c, y_c$ | Center coordinates of the ellipse | $[-L/8, L/8]$ |
| $x_0, y_0$ | Center of Gaussian conductivity distribution | $[0, 0.5]$ mm |
| $\sigma_0$ | Amplitude of Gaussian conductivity | $[2 \times 10^{-3}, 2.2 \times 10^{-2}]$ |
| $s$ | Variance of Gaussian conductivity distribution | $[0.2, 1]$ |
| $V_0$ | Electric potential of inner electrode | 70 V |
| $\varepsilon_p$ | Porosity | 0.6 |
| $a_{\text{pore}}$ | Average pore radius | 10 $\mu$m |
| $\mu$ | Dynamic viscosity of fluid | 0.001 Pa·s |
| $\varepsilon_w$ | Permittivity of fluid | $7.1 \times 10^{-11}$ F/m |
| $\zeta$ | Zeta potential | $-0.1$ V |
| $p_0$ | Outlet pressure | 1013 Pa |

**Magneto-Hydrodynamic (MHD) Coupling**

The basic electromagnetic–fluid coupling model is illustrated in Figure 6, where (a) and (b) show cross-sectional views of the electromagnetic and fluid models, respectively. The computational

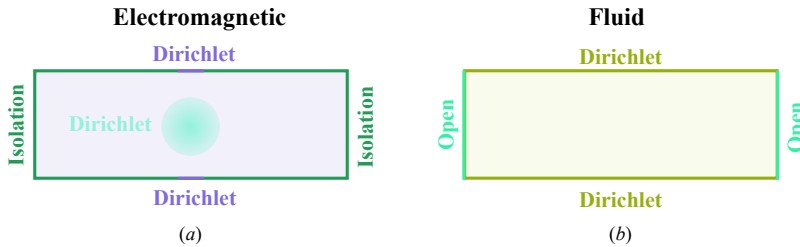

Figure 6: Simulation model of the electromagnetic–fluid coupling.

domain is a rectangular box with dimensions $L = 8$ cm, $W = 2.75$ cm, and $H = 2$ cm. For the electric field, Dirichlet boundary conditions (prescribed potential $V$ are applied at the electrode locations, while all other boundaries are electrically insulated:

$$V = V_0, y = -\frac{W}{2} \text{ and } x^2 + z^2 = r_1^2, \tag{29}$$

$$V = 0, y = \frac{W}{2} \text{ and } x^2 + z^2 = r_1^2, \tag{30}$$

$$\mathbf{n} \cdot \mathbf{J} = 0, \text{Others.} \tag{31}$$

Here, $V_0$ is the electrode potential, $r_1$ is the electrode radius, and $\mathbf{n}$ denotes the outward normal vector. For the magnetic field, Dirichlet boundary conditions are applied at the top and bottom surfaces, while the remaining boundaries are magnetically insulated:

$$\mathbf{B} = \mathbf{z}B_0 e^{-\frac{(x-x_0)^2+(y-y_0)^2}{2s^2}}, z = \pm H/2, \tag{32}$$

$$\mathbf{n} \times \mathbf{A} = 0. \tag{33}$$

Here, $B_0$, $x_0/y_0$, and $s$ denote the amplitude, center location, and variance of the magnetic field distribution, respectively; $\hat{z}$ is the unit vector in the $z$ direction, and $\mathbf{A}$ is the magnetic vector potential. For the fluid model, the left and right boundaries ($x = \pm L/2$) are treated as open boundaries, while all other surfaces are specified with no-slip boundary conditions:

$$\mathbf{n} \cdot \boldsymbol{\tau} - p\mathbf{n} = 0, x = \pm L/2, \tag{34}$$

$$\mathbf{u} = 0, \text{Others.} \tag{35}$$

Here, $\boldsymbol{\tau}$ denotes the stress tensor. Since the velocity field involves a body force related to the current density, and the current density itself contains an induced current component that depends on the velocity, the two fields are bidirectionally coupled.

After constructing the two physical models, we generated a total of 11,000 specimens using the finite element method. The corresponding parameter settings are listed in Table 8. The two components of the velocity field $(u_x, v_y)$, together with the three components of the current density $(J_x, J_y, J_z)$, are discretized into five separate $128 \times 128$ matrices, which serve as the learning targets for the neural network.

**Acoustic–Structure Coupling**

The fundamental model of acoustic–solid coupling is depicted in Figure 7, where subfigures (a) and (b) represent the acoustic and structural vibration models, respectively. A circular aluminum pillar with a radius of $r = 5\,\text{mm}$, centered at the origin, is embedded within a square domain of side length $L = 40\,\text{mm}$. The domain is filled with an inhomogeneous fluid whose density varies spatially according to:

$$\rho_c = 10 + A e^{-\frac{(x-x_0)^2+(y-y_0)^2}{2s^2}}, \tag{36}$$

Table 8: Parameters of the Magneto-Hydrodynami (MHD) Coupling Model

| Symbol | Physical Quantity | Value / Range |
|---|---|---|
| $L$ | Domain length | 8 cm |
| $W$ | Domain width | 2.75 cm |
| $H$ | Domain height | 2 cm |
| $r_1$ | Electrode radius | $H/6$ |
| $x_0, y_0$ | Center of Gaussian magnetic field | [–0.01, 0.01] |
| $B_0$ | Amplitude of magnetic field | [0, 1] T |
| $s$ | Variance of Gaussian magnetic field | [0.2, 1] |
| $V$ | Electrode potential | 0.1 V |
| $\mu_s$ | Magnetic permeability | $4\pi \times 10^{-7}$ H/m |
| $\sigma$ | Electrical conductivity of fluid | 10 S/m |
| $\mu$ | Dynamic viscosity of fluid | 0.001 Pa·s |
| $\rho$ | Fluid density | 1000 kg/m$^3$ |

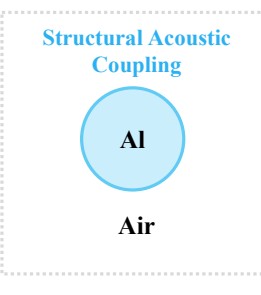

Figure 7: Simulation model of the Acoustic–Structure coupling.

where $A$, $x_0/y_0$, and $s$ denote the amplitude, center coordinates, and spatial spread of the density distribution, respectively. For the acoustic boundary conditions, the outer surface of the domain satisfies a scattering boundary condition given by:

$$\frac{\partial p}{\partial \mathbf{n}} = jkp, \tag{37}$$

where $\mathbf{n}$ is the unit normal vector directed outward from the boundary, and $k$ is the acoustic wave number. The inner boundary, representing the fluid–structure interface, satisfies the acoustic–structure coupling condition:

$$\mathbf{n} \cdot \left( \frac{1}{\rho_0} \nabla p \right) = (\mathbf{n} \cdot \mathbf{x})\, \omega^2, \tag{38}$$

where $\omega$ is the angular frequency. The mechanical boundary condition, applied only at the aluminum–fluid interface, is of Neumann type. The vibration-induced boundary load $\mathbf{F}$ is defined as the normal component of the fluid pressure acting on the solid. This model captures the bidirectional coupling between the acoustic and structural domains: the fluid pressure field exerts forces on the solid, inducing vibrations, while the resulting structural displacements in turn generate new acoustic waves.

Following the construction of this coupled physical model, a dataset comprising 11,000 samples was generated using the finite element method. The corresponding simulation parameters are summarized in Table 9. The outputs targeted by the neural network include six fields discretized into $128 \times 128$ matrices: two components of the displacement field ($u_x$ and $u_y$), three components of the stress tensor ($\sigma_{xx}$, $\sigma_{xy}$, and $\sigma_{yy}$), and the acoustic pressure field.

Table 9: Parameters of the Acoustic–Structure Coupling Model

| Symbol and | Physical Quantity | Value / Range |
|---|---|---|
| $L$ | Side length of the domain | 40 mm |
| $r$ | Radius of the circular region | 5 mm |
| $x_0, y_0$ | Center of the Gaussian source | [-0.01, 0.01] |
| $A$ | Amplitude of the Gaussian source | [500, 900] |
| $s$ | Variance of the Gaussian profile | [5, 20] |
| $k$ | Acoustic wave number | 210 rad/m |
| $\rho$ | Density of the solid | 2730 kg/m$^3$ |
| $c_c$ | Speed of sound in fluid | 1490 m/s |

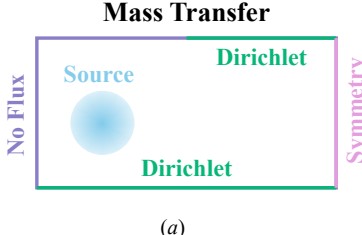

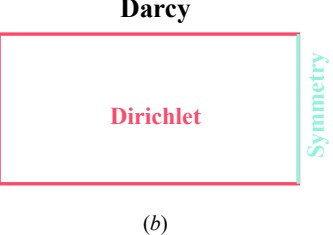

(a)          (b)

Figure 8: Simulation model of the Mass Transport–Fluid coupling. (a) Mass transport model; (b) Darcy flow model.

**Mass Transport–Fluid Coupling** The basic model of Mass Transport–Fluid coupling is illustrated in Figure 8, where (a) and (b) correspond to the mass transport model and the Darcy flow model, respectively. The computational domain is a rectangular region with length $2L$ and width $L$. For the mass transport problem, the boundary conditions for concentration $c$ are defined as follows: the bottom boundary and part of the top boundary are assigned Dirichlet conditions, the right boundary is symmetric, while the left and the remaining part of the top boundary are no-flux boundaries:

$$c = 0, y = 0, \tag{39}$$

$$c = c_s, x \in [L, 2L] \ and \ y = L, \tag{40}$$

$$\mathbf{n} \cdot [c\mathbf{u} - \theta_s \tau D_L \nabla c] = 0, x = 0 \ or \ x = 2L \ or, x \in [0, L] \ and \ y = L. \tag{41}$$

Here, $c_s$ is the boundary concentration, $\mathbf{u}$ is the unit outward normal vector, $\theta_s$, $\tau$, and $D_L$ denote the fluid volume fraction, tortuosity, and diffusion coefficient, respectively. Additionally, an internal mass source term $S_c$ is defined with a Gaussian profile:

$$S_c = \frac{A}{T_0} e^{-\frac{(x-x_0)^2 + (y-y_0)^2}{2s^2}}, \tag{42}$$

where $A$, $x_0/y_0$, and $s$ describe the amplitude, center, and standard deviation of the Gaussian source, and $T_0$ is a temporal scaling constant. The initial condition for the concentration field is:

$$c = 0, t = 0. \tag{43}$$

For the Darcy flow problem, the right boundary is assigned a symmetry condition, and all other boundaries are specified as no-flow:

$$\frac{\partial \rho \mathbf{u}}{\partial x} = 0, x = 2L, \tag{44}$$

$$\mathbf{n} \cdot \rho \mathbf{u} = 0, \text{Others}. \tag{45}$$

The initial condition for the pressure field is:

$$p(x, y, 0) = \rho_0 \mathbf{g} \cdot \mathbf{D}, \tag{46}$$

where $\rho_0$ is the density of pure water, $\mathbf{g}$ is the gravitational acceleration vector, and $\mathbf{D}$ is the elevation vector. Clearly, the Darcy velocity $\mathbf{u}$ influences the concentration distribution, while concentration gradients also affect the flow velocity, indicating a bidirectional coupling between the two fields. After constructing the physical model, we generated 11,000 datasets using the finite element method. The simulation covers a time span from 0 to 20 years, with a sampling interval of 2 years. Table 10 presents the geometric and physical parameters used in the Elder problem. The computed transient velocity field components ($u_x$ and $u_y$) and the concentration field $c$ are discretized into three $11 \times 128 \times 128$ tensors, which serve as learning targets for the neural network.

Table 10: Parameters of the Mass Transport–Fluid Coupling Model

| Symbol | Physical Quantity | Value / Range |
|--------|-------------------|---------------|
| $L$ | Domain width | 150 m |
| $c_s$ | Boundary concentration | 1 mol/m$^3$ |
| $\beta$ | Density–concentration coefficient | 200 kg/mol |
| $x_0, y_0$ | Center of Gaussian source | [0, 0.5] |
| $\sigma_0$ | Amplitude of Gaussian source | [0, 0.05] |
| $s$ | Variance of Gaussian source | [0.2, 1] |
| $\varepsilon_p$ | Porosity | 0.1 |
| $\tau D_L$ | Molecular diffusion coefficient | $3.56 \times 10^{-6}$ m$^2$/s |
| $\mu$ | Dynamic viscosity of fluid | 0.001 Pa·s |
| $\theta_s$ | Fluid volume fraction | 0.1 |

**Thermo-Fluid-Solid Coupling** (Energy, Navier–Stokes, and Structural Vibration Equation) This model describes the deformation process induced by aerodynamic heating in high–speed fluid flows. The key physical variables include the temperature $T$, the velocity field $\mathbf{u}/v$, and the solid displacement $x/y$. When exposed to a high–temperature fluid, the metallic sheet undergoes thermal expansion, leading to structural deformation. The governing equations of the coupled system are:

$$\rho C_p \frac{\partial T}{\partial t} + \rho C_p \mathbf{u} \cdot \nabla T - \nabla \cdot (\kappa \nabla T) = Q, \tag{47}$$

$$\rho C_p \frac{\partial \mathbf{u}}{\partial t} + \rho \left( \mathbf{u} \cdot \nabla \right) \mathbf{u} = -\nabla p + \mu \nabla^2 \mathbf{u} + \rho \mathbf{g}, \rho_s \frac{\partial^2 \mathbf{x}}{\partial t^2} = \nabla \cdot (\mathbf{FS})^T + \mathbf{F}_v. \tag{48}$$

Here, $\rho$ and $\rho_s$ denote the fluid and solid densities, respectively; $C_p$ is the specific heat capacity at constant pressure; $\kappa$ is the thermal conductivity; $p$ is the pressure; $\mu$ is the dynamic viscosity; and $\mathbf{g}$ is the gravitational acceleration. $\mathbf{F}$ and $\mathbf{S}$ represent the deformation gradient and the second Piola–Kirchhoff stress tensor, respectively, while $\mathbf{F}_v$ denotes the volumetric body force. This thermo–fluid–solid coupling mechanism has been extensively applied in diverse fields, including turbine blade cooling, thermal protection systems for spacecraft, and MEMS–based thermal sensors.

The thermo–fluid–solid coupling model is illustrated in Fig. 9, where panels (a)–(c) respectively represent the heat transfer, fluid, and solid domains. The computational domain is a rectangular channel of length $l_c$ and width $h_c$, containing a metallic plate of length $l_s$ and width $h_s$. For the heat transfer problem, the upper and lower boundaries are assumed to be adiabatic, i.e.,

$$-\mathbf{n} \cdot \mathbf{q} = 0, \tag{49}$$

where $\mathbf{q}$ is the heat flux vector and $\mathbf{n}$ is the outward unit normal vector. The left and right boundaries are subject to flux conditions,

$$-\mathbf{n} \cdot \mathbf{q} = \rho C_p \mathbf{u} \cdot \mathbf{n}, \tag{50}$$

where $\rho$ denotes the fluid density, $C_p$ the specific heat capacity at constant pressure, and $\mathbf{u}$ the velocity vector. The metallic plate introduces a volumetric heat source given by

$$Q = \frac{Q_A}{l_s h_s} \left[ (x + Q_x)^2 + 10 \left( y + Q_y \right)^2 \right], \tag{51}$$

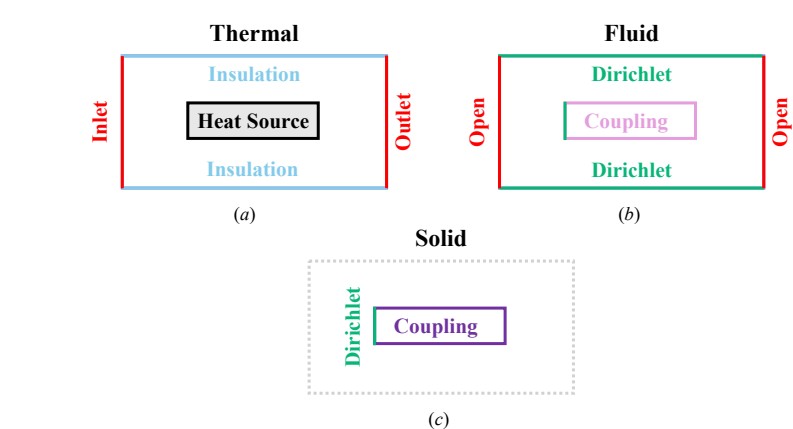

Figure 9: Simulation model of the Thermo-Fluid-Solid coupling.

where the parameter $Q_A$ defines the source intensity, while $Q_x$ and $Q_y$ control its spatial location. The initial temperature field is specified as

$$T(x, y, 0) = T_0. \tag{52}$$

For the fluid domain, the upper and lower channel boundaries, as well as the left boundary of the metallic plate, are subject to no-slip Dirichlet conditions,

$$\mathbf{u} = \mathbf{0}. \tag{53}$$

The left and right channel boundaries are treated as open boundaries,

$$\mathbf{u} \cdot \mathbf{t} = 0, \tag{54}$$

where $\mathbf{t}$ is the tangential unit vector at the boundary. The remaining boundaries of the metallic plate are subject to coupling conditions,

$$\mathbf{u}_{\text{fluid}} = \mathbf{u}_{\text{solid}} \tag{55}$$

The initial velocity field is given by

$$\mathbf{u}(x, y, 0) = \mathbf{0}. \tag{56}$$

For the solid domain, boundary conditions are applied only at the fluid–solid interface. The left boundary is prescribed with a Dirichlet condition,

$$\mathbf{x} = \mathbf{0}, \tag{57}$$

while the other boundaries satisfy fluid–solid coupling conditions,

$$\mathbf{x}_{\text{fluid}} = \mathbf{x}_{\text{solid}}. \tag{58}$$

The initial displacement field is defined as

$$\mathbf{x}(x, y, 0) = \mathbf{0}. \tag{59}$$

Within this multiphysics framework, heat transfer and fluid flow are bidirectionally coupled, fluid and solid are also bidirectionally coupled, whereas the heat transfer–solid interaction is unidirectional. Based on the physical model, we generated 1000 samples using the finite element method. The simulations span a temporal range of 0–3000 s with a sampling interval of 600 s. Table 11 summarizes the geometric and physical parameters employed in the Elder problem. The computed transient fields—temperature, velocity components ($u_x$, $u_y$), and displacement components ($x_x$, $y_y$)—are discretized into three tensors of dimension ($5 \times 128 \times 128$), which serve as the learning targets for the neural network.

**Schrödinger–Poisson Coupling** (Time-Dependent Schrödinger Equation and Poisson Equation) This model describes the evolution of a quantum wave packet under a self-consistent electrostatic

Table 11: Parameters of the Thermo-Fluid-Solid Coupling Model

| Symbol | Physical Quantity | Value / Range |
|--------|-------------------|---------------|
| $l_c$ | Domain length | 0.7 m |
| $h_c$ | Domain width | 0.1 m |
| $l_s$ | Metallic plate length | 0.1 m |
| $h_s$ | Metallic plate width | 0.01 |
| $\rho$ | Fluid density | 1.293 kg/m$^3$ |
| $C_p$ | Heat Capacity | 1005J/(kg· K) |
| $Q_A$ | Amplitude coefficient | [3000,3500] |
| $Q_x$ | $x$ Center of the Gaussian source | [-0.02,0.02] |
| $Q_y$ | $y$ Center of the Gaussian source | [-0.02,0.02] |

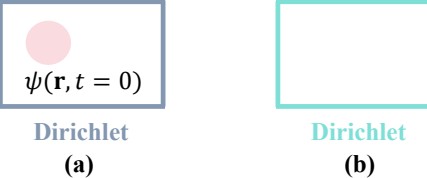

**Probability Amplitude**   **Electrostatic Potential**

$\psi(\mathbf{r}, t = 0)$

Dirichlet         Dirichlet

**(a)**              **(b)**

Figure 10: Simulation model of the Schrödinger–Poisson coupling.

potential. The key physical variables include the complex wave function $\psi$, representing the quantum probability amplitude, and the electric potential $\phi$ generated by the charge density $|\psi|^2$. The system forms a bidirectionally coupled quantum–electrostatic model: the potential $\phi$ influences the quantum dynamics through the Hamiltonian, while the wave function $\psi$ simultaneously serves as the source term for the Poisson equation. The governing equations of the coupled system are:

$$i\hbar \frac{\partial \psi(\mathbf{r}, t)}{\partial t} = \left[ -\frac{\hbar^2}{2m} \nabla^2 - e\, \phi(\mathbf{r}, t) \right] \psi(\mathbf{r}, t), \tag{60}$$

$$\varepsilon \nabla^2 \phi(\mathbf{r}, t) = e\, |\psi(\mathbf{r}, t)|^2. \tag{61}$$

Here, $\hbar$ is the reduced Planck constant, $m$ is the particle mass, $e$ is the elementary charge, and $\varepsilon$ denotes the permittivity of the material. The first equation governs the time evolution of the quantum wave function, while the second equation computes the electrostatic potential generated by the particle density $|\psi(\mathbf{r}, t)|^2$. The Schrödinger–Poisson coupling is widely used in quantum device modeling, including semiconductor nanostructures, quantum wells, and nanoscale transistors, where self-consistent charge–potential interactions are essential. It also plays a central role in simulating quantum transport and carrier confinement in modern electronic and optoelectronic devices.

The Schrödinger–Poisson coupling model is illustrated in Fig. 10, where panel (a) shows the evolution of the wave function amplitude and panel (b) shows the electrostatic potential. The computational domain is a square region of size $L \times L$. Homogeneous Dirichlet boundary conditions are imposed on all four edges,

$$\psi = 0, \phi = 0, \tag{62}$$

corresponding to a quantum well with grounded boundaries. The initial quantum state is prescribed as

$$\psi(x, y, 0) = \psi_0(x, y), \tag{63}$$

where $\psi_0$ is typically chosen as a Gaussian wave packet or a low-lying eigenmode of the unperturbed Hamiltonian. Within this multiphysics framework, the Schrödinger and Poisson equations are fully bidirectionally coupled: the instantaneous electrostatic potential modifies the quantum wavefunction

evolution, while the Poisson equation is solved at every time step using the charge density $|\psi(\mathbf{r}, t)|^2$ generated from the updated wavefunction. To construct the dataset, we numerically solved the coupled Schrödinger–Poisson system for 1000 independently sampled initial Gaussian wave packets. The Schrödinger equation is discretized using a Crank–Nicolson semi-implicit scheme, where the kinetic term is represented by a sparse 2D Laplacian operator and the potential term is updated self-consistently at each step. The Poisson equation is solved on the same grid by directly inverting the discrete Laplacian for the interior nodes under Dirichlet boundary conditions. Each simulation spans a temporal window $[0, T_{\max}]$ with a uniform step size $\Delta t$. The outputs is the real and imaginary parts of $\psi$, together with the potential field $\phi$, which are stored as tensors of size $(N_t \times N_x \times N_y)$ and serve as supervised training targets for neural networks.

## A.2 DATASET COLLECTION

The datasets utilized in this study were generated via numerical simulations using a licensed version of *COMSOL Multiphysics*, integrated with *MATLAB* through the *COMSOL LiveLink* interface. The overall workflow begins with the identification of relevant physical problems and their corresponding governing equations, including multiphysics coupling mechanisms. These problems are solved using the finite element method (FEM), encompassing the design of appropriate geometries, definition of physical fields and coupling relationships, specification of material properties, mesh generation, solver configuration, numerical computation, and post-processing.

Upon completion of the simulations, each COMSOL model is exported as a MATLAB function (*.m file) via LiveLink. These functions accept input parameters and return the corresponding physical field values. Automated *MATLAB* scripts are then employed to sweep across the defined parameter space, facilitating the generation of diverse datasets for training purposes. All relevant source files will be made publicly available as part of our open-source dataset release. The software stack employed in this process is institutionally licensed and fully compliant with usage guidelines.

**Electro-Thermal Coupling:** The input comprises a single channel encoding electrical conductivity, thermal conductivity, and geometric information. The output consists of three channels: $\text{Re}(E_z)$, $\text{Im}(E_z)$, and the temperature field $T$.

**Thermo-Fluid Coupling:** The input consists of a spatially varying heat source and its spatial position (1 channel), while the output includes the temperature field and a 2D velocity field (3 channels total).

**Electro-Fluid Coupling:** The input is a Gaussian-distributed conductivity field combined with variable geometry (1 channel). The outputs include the electric potential and a 2D velocity field (3 channels total).

**Magneto-Hydrodynamic (MHD) Coupling:** The input is a single channel representing a randomly sampled remanent magnetic flux. The outputs include current density (2 channels) and velocity components (3 channels), yielding a total of 5 output channels.

**Acoustic–Structure Coupling:** The input is the spatial material density (1 channel). The outputs comprise the acoustic pressure field (2 channels), stress components (6 channels), and structural displacements (4 channels), for a total of 12 output channels.

**Mass Transport–Fluid Coupling:** The input includes the source term $S_c$ and the initial state of the system at time $t_0$ (concentration and velocity), totaling 4 channels. The output consists of the predicted concentration and velocity fields across 10 future time steps (30 channels in total).

## B EXPERIMENTAL DETAILS

### B.1 METRICS

To comprehensively evaluate model performance across different multiphysics scenarios, we adopt the evaluation metrics proposed in PDEBench (Takamoto et al., 2022), as summarized in Table 12.

In particular, for the frequency-domain Root Mean Squared Error (fRMSE), we define three distinct frequency bands—low, middle, and high—to facilitate fine-grained analysis of prediction accuracy across different spectral regions. The fRMSE for each band is defined as:

$$\text{fRMSE}^{(m)} = \frac{1}{|F_m|}\sqrt{\sum_{k \in F_m} |\mathcal{F}(\hat{y})_k - \mathcal{F}(y)_k|^2}, \quad m \in \{\text{low}, \text{middle}, \text{high}\}, \tag{64}$$

where $\mathcal{F}(\cdot)$ denotes the discrete Fourier transform, and $F_{\text{low}} = [0, 4]$, $F_{\text{middle}} = [5, 12]$, and $F_{\text{high}} = [13, N//2]$, with $N$ representing the discrete signal length in the frequency domain.

Table 12: Definitions and Interpretations of Evaluation Metrics

| Metric | Expression | Interpretation |
|---|---|---|
| RMSE | $\sqrt{\frac{1}{n}\sum_{i=1}^{n}(\hat{y}_i - y_i)^2}$ | Measures the standard deviation of the prediction residuals at all data points. |
| Relative $L_2$ Error | $\frac{\|\hat{y}-y\|_2}{\|y\|_2}$ | Normalized error magnitude showing deviation as a fraction of reference values. |
| Max Error | $\max_i |\hat{y}_i - y_i|$ | Worst-case pointwise prediction error, critical for safety-sensitive applications. |
| bRMSE | $\sqrt{\frac{1}{n_b}\sum_{j \in \Omega_b}(\hat{y}_j - y_j)^2}$ | Specialized RMSE evaluating boundary condition accuracy ($\Omega_b$: boundary point set). |
| fRMSE | $\sqrt{\frac{1}{K}\sum_{k=1}^{K}|\mathcal{F}(\hat{y})_k - \mathcal{F}(y)_k|^2}$ | Spectral accuracy metric via DFT ($\mathcal{F}$: Fourier transform operator). |

## B.2 IMPLEMENTATION CONFIGURATION

In our experimental setup, the dataset for the Mass Transport–Fluid transient problems each contain 1,000 training samples and 100 validation samples. All other multiphysics tasks use datasets of 10,000 training and 1,000 validation samples to enable consistent evaluation. The training configurations are summarized in Table 13. For Physics-Informed Neural Networks (PINNs), we use a residual-block backbone (layer_num = 2; hidden_dim $\in \{128, 256\}$) with separate output heads for each physical field, and train with a physics-informed loss whose PDE and data terms are initially weighted 1:1000. To mitigate imbalance among multiple PDE losses, we apply an adaptive reweighting scheme that normalizes loss magnitudes and stabilizes training. DeepONet adopts an unstacked design: a four-layer convolutional BranchNet (hidden_dim $\in \{128, 256\}$) and a two-layer MLP TrunkNet. For DiffusionPDE, we employ observation-based guidance restricted to the initial state only; no final-state information is used to ensure a fair comparison across methods.

Table 13: Training configurations for different baselines.

| Model | Batch Size | Learning Rate | Epochs | Optimizer |
|---|---|---|---|---|
| PINNs | 48 | $5 \times 10^{-3}$ | 50 | Adam |
| DeepONet | 48 | $4 \times 10^{-3}$ | 250 | Adam |
| FNO | 48 | $1 \times 10^{-4}$ | 50 | Adam |
| DiffusionPDE | 96 | $1 \times 10^{-3}$ | 100 | Adam |

## C SUPPLEMENTARY EMPIRICAL RESULTS

### C.1 DETAILED BASELINE PERFORMANCE

In this section, we evaluate the performance of the baseline model across various physical problems. Specifically, Figures 15-20 present additional multiphysics field samples that serve as a supplementary dataset to the information provided in Figure 2. We present the probability density distributions

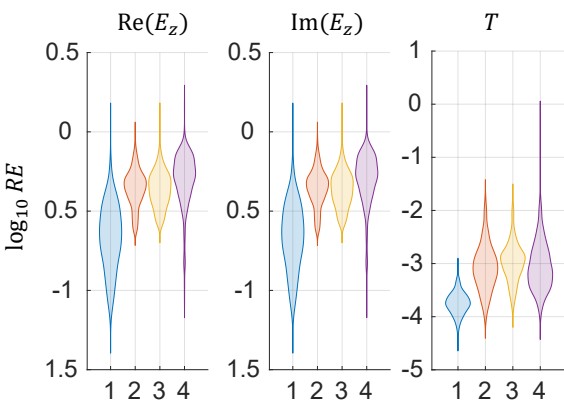

Figure 11: Relative $L_2$ error distribution on the Electro-Thermal dataset. 1-4 indicate DeepONet, FNO, PINNs and DiffusionPDE.

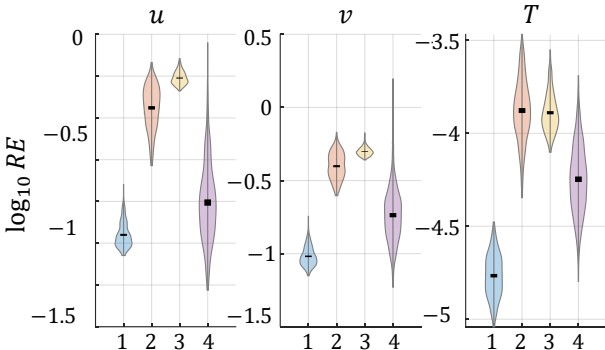

Figure 12: Relative $L_2$ error distribution on the Thermo-Fluid dataset. 1-4 indicate DeepONet, FNO, PINNs and DiffusionPDE, and the black regions represent the $95\%$ confidence intervals.

of the relative $L_2$ errors on the test set for the Electro-Thermal coupling task across four SciML models in Figure 11. Overall, DeepONet achieves the lowest median error, while DiffusionPDE exhibits noticeable long-tail behavior in its error distribution. In addition, the training loss of the FNO network for the Electro-Thermal coupling task and the Thermo-Fluid task is presented in 23

**Future Directions**   While Multiphysics Bench offers a comprehensive suite of real-world multi-physics problems with standardized implementations and detailed analyses, several important chal-lenges remain: (1) extending the benchmark to encompass a broader range of multiphysics scenarios, particularly time-dependent systems with increased scale and complexity, (2) developing targeted improvements based on baseline limitations, including the incorporation of explicit cross-physics coupling mechanisms and effective physics-informed supervisory strategies, and (3) designing mul-tiphysics foundation models with data, compute, and model size scaling laws.

## C.2   DETAILED COMPLETE PHYSICAL PRIOR VERSUS INCOMPLETE PHYSICAL PRIOR

To complement the results in Table 11, we conduct a comparative experiment to evaluate the influ-ence of complete versus incomplete physical priors on model performance. Specifically, we focus on the Electro-Thermal multiphysics dataset, where we assume that only the electric field is observ-able during training. The corresponding predictive results under this partially supervised setting are reported in Table 21.

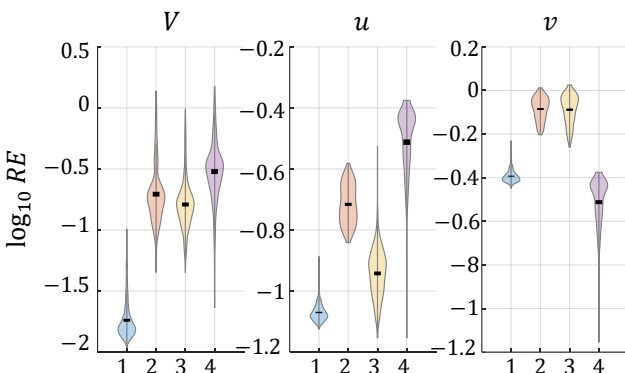

Figure 13: Relative $L_2$ error distribution on the Electro-Fluid dataset. 1-4 indicate DeepONet, FNO, PINNs and DiffusionPDE, and the black regions represent the 95% confidence intervals.

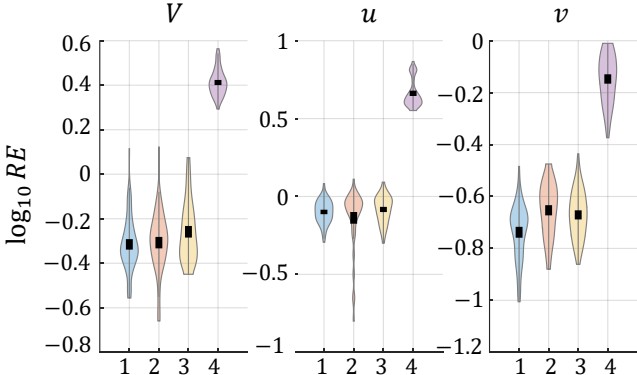

Figure 14: Relative $L_2$ error distribution on the Mass Transport–Fluid dataset at time step $t = 10$. 1-4 indicate DeepONet, FNO, PINNs and DiffusionPDE, and the black regions represent the 95% confidence intervals.

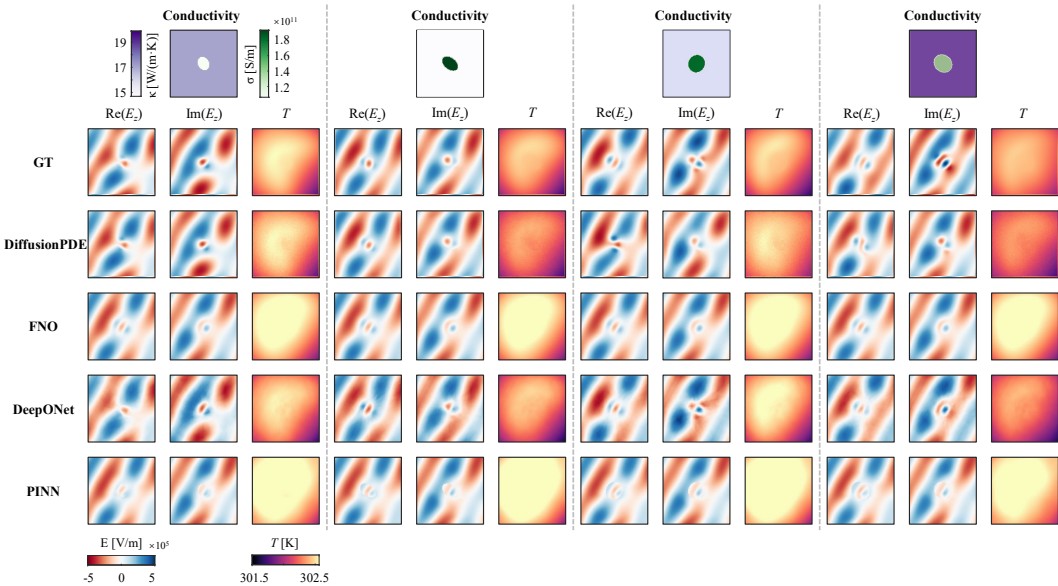

Figure 15: The additional baseline result of the Electro-Thermal Coupling problem. The inputs are the distribution of the electrical conductivity $\sigma$ and thermal conductivity $\kappa$, while the outputs are the complex electric field $E_z$ and temperature $T$.

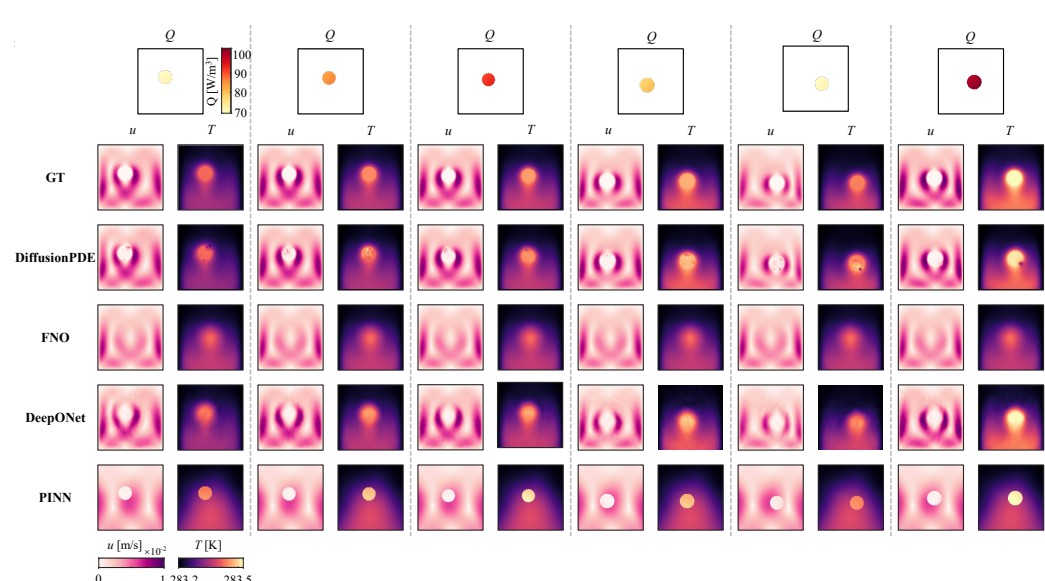

Figure 16: The additional baseline result of the Thermo-Fluid Coupling problem. The inputs are the distribution of the heat source density $Q$, while the outputs are the temperature $T$ and velocity field $\mathbf{u}$ at $x$ and $y$ direction.

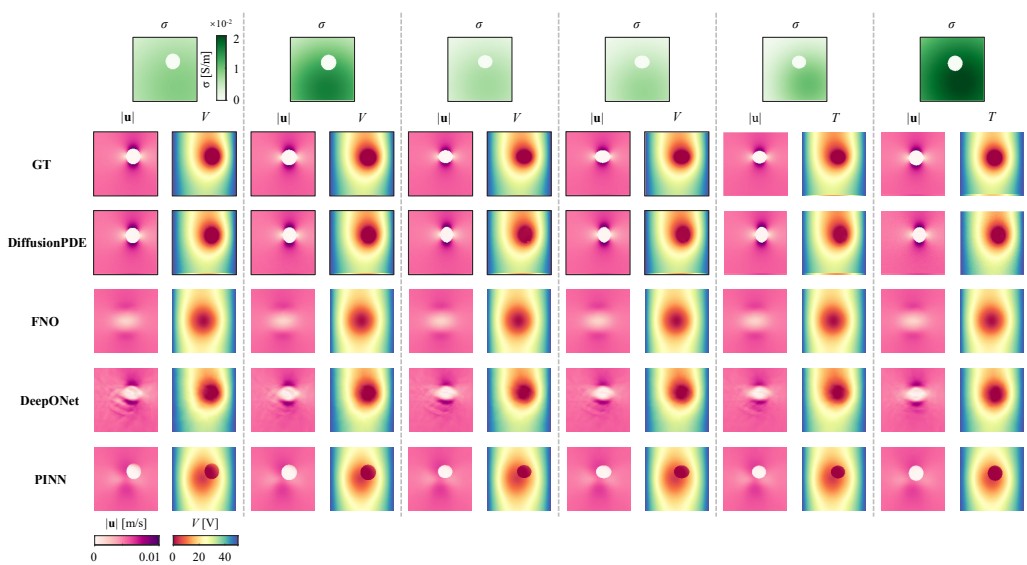

Figure 17: The additional baseline result of the Electro-Fluid problem. The inputs are the distribution of the conductivity $\rho$, while the outputs are the electrical potential $V$ and velocity field $\mathbf{u}$ at $x$ and $y$ direction.

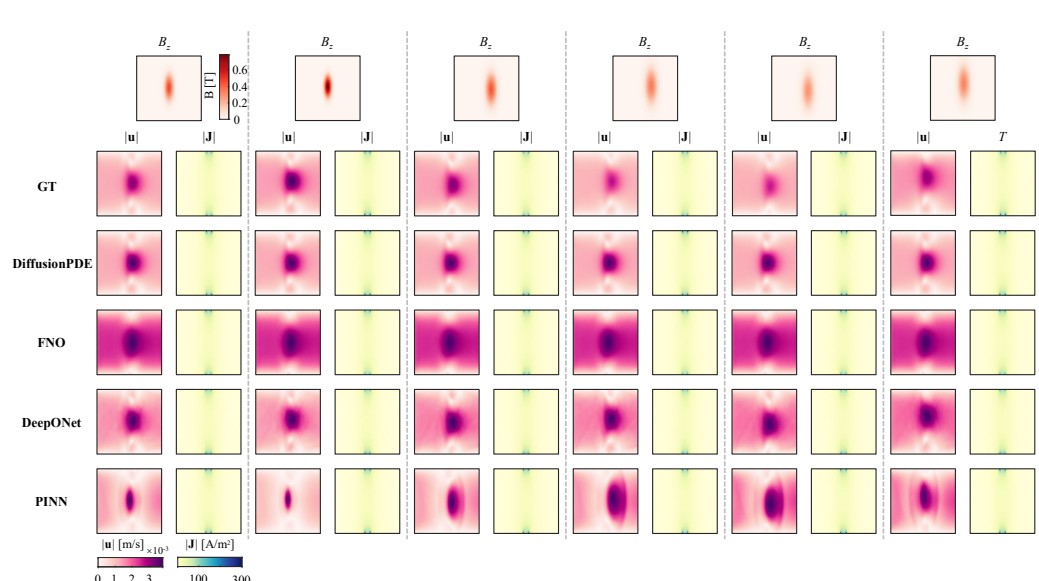

Figure 18: The additional baseline result of the Magneto-Hydrodynamic (MHD) problem. The inputs are the distribution of the external magnetic field $B_z$, while the outputs are the electrical current density $\mathbf{J}$ and velocity field $\mathbf{u}$ at $x, y$ and $z$ direction.

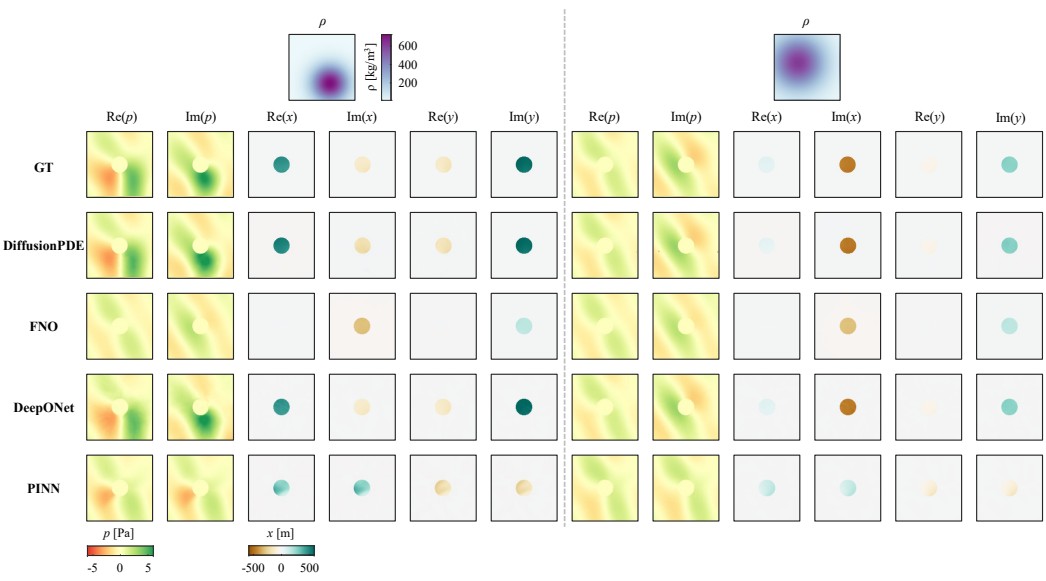

Figure 19: The additional baseline result of the Acoustic–Structure Coupling Problems. The inputs are the density $\rho$, while the outputs are the complex pressure $p$ and displacement $\mathbf{x}$ at $x$ and $y$ direction. All displacement values are scaled by $10^6$ for numerical convenience.

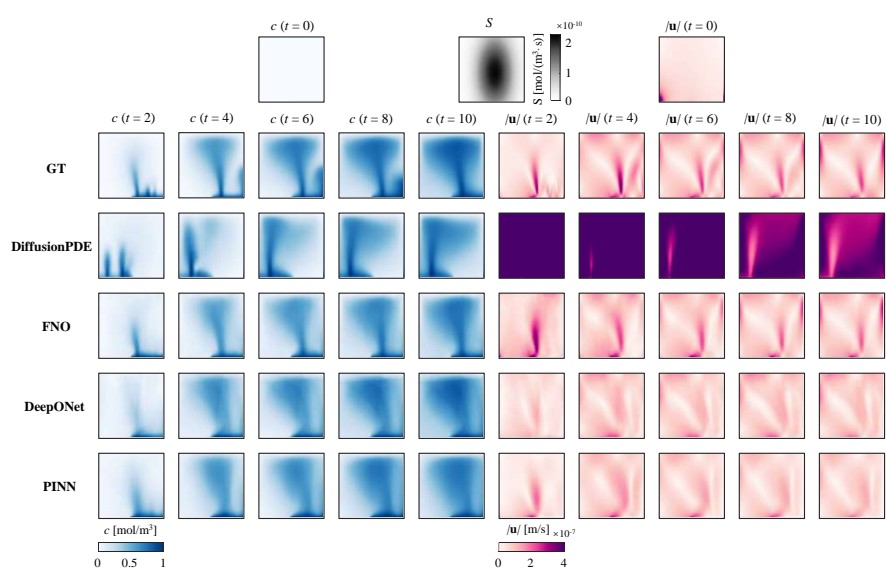

Figure 20: The additional baseline result of the Mass Transport–fluid Coupling Problems. The inputs are the source density $S$ and , while the outputs are the complex pressure $p$ and displacement $\mathbf{x}$ at $x$ and $y$ direction.

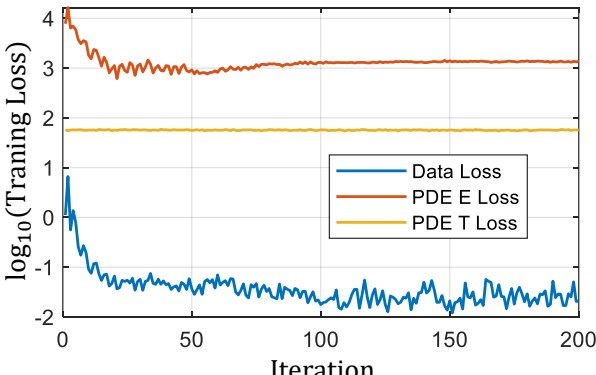

Figure 21: Training Loss of the FNO Model for the Electro-Thermal Task.

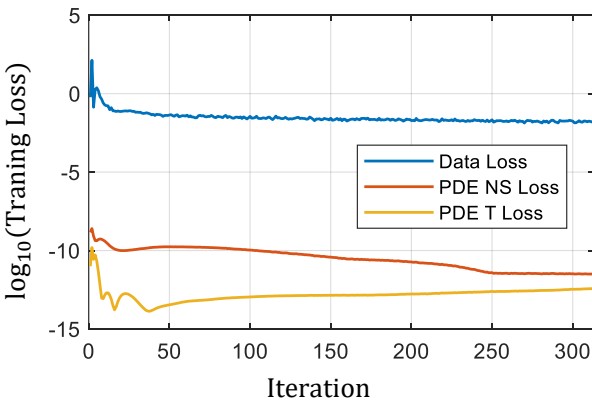

Figure 22: Training Loss of the FNO Model for the Thermo-Fluid Task.

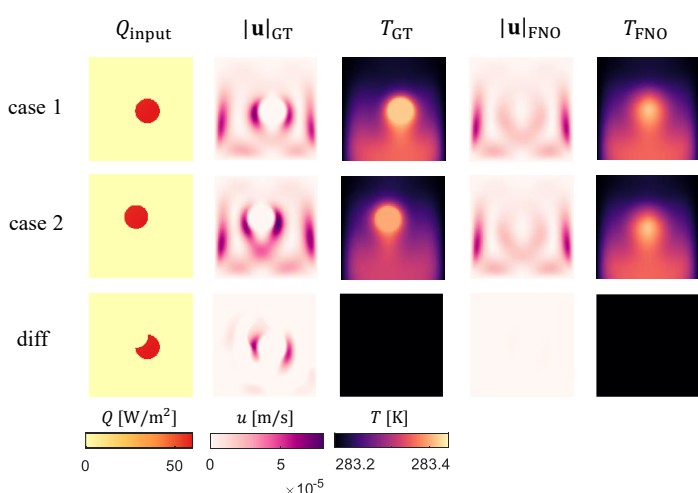

Figure 23: Visualization of mode collapse in FNO.

Table 14: Baseline model performance on the Electro-Thermal task under different evaluation metrics.

| Metric | Fields | PINNs | DeepONet | FNO | DiffusionPDE |
|---|---|---|---|---|---|
| RMSE | $\mathrm{Re}(E_z)$ | $9.197 \times 10^4$ | $\mathbf{5.420 \times 10^4}$ | $9.209 \times 10^4$ | $1.152 \times 10^5$ |
| | $\mathrm{Im}(E_z)$ | $9.324 \times 10^4$ | $\mathbf{5.461 \times 10^4}$ | $9.343 \times 10^4$ | $1.107 \times 10^5$ |
| | $T$ | $4.772 \times 10^{-1}$ | $7.026 \times 10^{-1}$ | $\mathbf{4.500 \times 10^{-1}}$ | $4.737 \times 10^{-1}$ |
| Relative $L_2$ Error | $\mathrm{Re}(E_z)$ | $4.387 \times 10^{-1}$ | $\mathbf{2.573 \times 10^{-1}}$ | $4.388 \times 10^{-1}$ | $5.499 \times 10^{-1}$ |
| | $\mathrm{Im}(E_z)$ | $4.469 \times 10^{-1}$ | $\mathbf{2.617 \times 10^{-1}}$ | $4.474 \times 10^{-1}$ | $5.330 \times 10^{-1}$ |
| | $T$ | $1.573 \times 10^{-3}$ | $2.317 \times 10^{-3}$ | $\mathbf{1.482 \times 10^{-3}}$ | $1.560 \times 10^{-3}$ |
| MaxError | $\mathrm{Re}(E_z)$ | $3.989 \times 10^5$ | $\mathbf{2.621 \times 10^5}$ | $3.927 \times 10^5$ | $5.189 \times 10^5$ |
| | $\mathrm{Im}(E_z)$ | $4.017 \times 10^5$ | $\mathbf{2.666 \times 10^5}$ | $3.894 \times 10^5$ | $4.652 \times 10^5$ |
| | $T$ | $6.336 \times 10^{-1}$ | $8.583 \times 10^{-1}$ | $\mathbf{5.689 \times 10^{-1}}$ | $7.343 \times 10^{-1}$ |
| fRMSE low | $\mathrm{Re}(E_z)$ | $6.665 \times 10^6$ | $\mathbf{3.542 \times 10^6}$ | $6.715 \times 10^6$ | $2.384 \times 10^8$ |
| | $\mathrm{Im}(E_z)$ | $6.691 \times 10^6$ | $\mathbf{3.582 \times 10^6}$ | $6.643 \times 10^6$ | $2.307 \times 10^8$ |
| | $T$ | $5.873 \times 10^1$ | $7.488 \times 10^1$ | $\mathbf{5.795 \times 10^1}$ | $1.098 \times 10^3$ |
| fRMSE middle | $\mathrm{Re}(E_z)$ | $1.792 \times 10^6$ | $\mathbf{1.043 \times 10^6}$ | $1.811 \times 10^6$ | $3.622 \times 10^7$ |
| | $\mathrm{Im}(E_z)$ | $1.785 \times 10^6$ | $\mathbf{1.095 \times 10^6}$ | $1.757 \times 10^6$ | $3.335 \times 10^7$ |
| | $T$ | $7.816 \times 10^{-1}$ | $8.301 \times 10^{-1}$ | $\mathbf{5.916 \times 10^{-1}}$ | $1.520 \times 10^1$ |
| fRMSE high | $\mathrm{Re}(E_z)$ | $9.464 \times 10^5$ | $\mathbf{5.590 \times 10^5}$ | $9.517 \times 10^5$ | $1.722 \times 10^6$ |
| | $\mathrm{Im}(E_z)$ | $9.549 \times 10^5$ | $\mathbf{5.856 \times 10^5}$ | $9.510 \times 10^5$ | $1.728 \times 10^6$ |
| | $T$ | $5.783 \times 10^{-1}$ | $6.560 \times 10^{-1}$ | $\mathbf{4.988 \times 10^{-1}}$ | $4.187 \times 10^0$ |
| bRMSE | $\mathrm{Re}(E_z)$ | $6.333 \times 10^4$ | $\mathbf{3.664 \times 10^4}$ | $6.218 \times 10^4$ | $7.737 \times 10^4$ |
| | $\mathrm{Im}(E_z)$ | $6.330 \times 10^4$ | $\mathbf{3.587 \times 10^4}$ | $6.140 \times 10^4$ | $7.285 \times 10^4$ |
| | $T$ | $4.644 \times 10^{-1}$ | $6.804 \times 10^{-1}$ | $\mathbf{4.371 \times 10^{-1}}$ | $4.500 \times 10^{-1}$ |

This experiment illustrates that reducing a multiphysics problem to a single-physics formulation leads to the loss of essential cross-domain information and consequently degrades predictive accuracy. These findings underscore the importance of constructing and leveraging multiphysics datasets that retain the full spectrum of inter-physical interactions.

Table 15: Baseline model performance on the Thermo-Fluid task under different evaluation metrics.

| Metric | Fields | PINNs | DeepONet | FNO | DiffusionPDE |
|---|---|---|---|---|---|
| RMSE | $u$ | $1.139 \times 10^{-3}$ | $\mathbf{2.020 \times 10^{-4}}$ | $8.505 \times 10^{-4}$ | $3.730 \times 10^{-4}$ |
|  | $v$ | $1.600 \times 10^{-3}$ | $\mathbf{3.090 \times 10^{-4}}$ | $1.304 \times 10^{-3}$ | $6.130 \times 10^{-4}$ |
|  | $T$ | $3.762 \times 10^{-2}$ | $\mathbf{4.996 \times 10^{-3}}$ | $4.020 \times 10^{-2}$ | $3.012 \times 10^{-2}$ |
| Relative $L_2$ Error | $u$ | $6.204 \times 10^{-1}$ | $\mathbf{1.107 \times 10^{-1}}$ | $4.616 \times 10^{-1}$ | $1.999 \times 10^{-1}$ |
|  | $v$ | $5.020 \times 10^{-1}$ | $\mathbf{9.721 \times 10^{-2}}$ | $4.089 \times 10^{-1}$ | $1.889 \times 10^{-1}$ |
|  | $T$ | $1.330 \times 10^{-4}$ | $\mathbf{1.760 \times 10^{-5}}$ | $1.419 \times 10^{-4}$ | $1.060 \times 10^{-4}$ |
| MaxError | $u$ | $4.219 \times 10^{-3}$ | $\mathbf{1.706 \times 10^{-3}}$ | $3.505 \times 10^{-3}$ | $4.574 \times 10^{-3}$ |
|  | $v$ | $6.399 \times 10^{-3}$ | $\mathbf{3.018 \times 10^{-3}}$ | $5.579 \times 10^{-3}$ | $7.336 \times 10^{-3}$ |
|  | $T$ | $1.754 \times 10^{-1}$ | $4.414 \times 10^{-2}$ | $1.648 \times 10^{-1}$ | $\mathbf{7.336 \times 10^{-3}}$ |
| fRMSE low | $u$ | $7.133 \times 10^{-2}$ | $\mathbf{3.564 \times 10^{-3}}$ | $5.629 \times 10^{-2}$ | $6.566 \times 10^{-1}$ |
|  | $v$ | $1.109 \times 10^{-1}$ | $\mathbf{4.887 \times 10^{-3}}$ | $9.357 \times 10^{-2}$ | $1.038 \times 10^{0}$ |
|  | $T$ | $3.420 \times 10^{0}$ | $\mathbf{1.279 \times 10^{-1}}$ | $4.227 \times 10^{0}$ | $6.644 \times 10^{1}$ |
| fRMSE middle | $u$ | $1.645 \times 10^{-2}$ | $\mathbf{7.336 \times 10^{-3}}$ | $9.976 \times 10^{-3}$ | $1.374 \times 10^{-1}$ |
|  | $v$ | $3.973 \times 10^{-2}$ | $\mathbf{1.146 \times 10^{-2}}$ | $1.934 \times 10^{-2}$ | $2.542 \times 10^{-1}$ |
|  | $T$ | $5.758 \times 10^{-1}$ | $\mathbf{1.930 \times 10^{-1}}$ | $3.732 \times 10^{-1}$ | $5.313 \times 10^{0}$ |
| fRMSE high | $u$ | $1.207 \times 10^{-2}$ | $\mathbf{2.277 \times 10^{-3}}$ | $8.828 \times 10^{-3}$ | $1.707 \times 10^{-2}$ |
|  | $v$ | $1.602 \times 10^{-2}$ | $\mathbf{3.495 \times 10^{-3}}$ | $1.316 \times 10^{-2}$ | $2.793 \times 10^{-2}$ |
|  | $T$ | $3.474 \times 10^{-1}$ | $\mathbf{5.705 \times 10^{-2}}$ | $3.668 \times 10^{-1}$ | $6.468 \times 10^{-1}$ |
| bRMSE | $u$ | $1.497 \times 10^{-3}$ | $1.500 \times 10^{-4}$ | $\mathbf{6.106 \times 10^{-5}}$ | $9.800 \times 10^{-5}$ |
|  | $v$ | $2.925 \times 10^{-3}$ | $2.250 \times 10^{-4}$ | $\mathbf{7.262 \times 10^{-5}}$ | $1.720 \times 10^{-4}$ |
|  | $T$ | $4.542 \times 10^{-2}$ | $\mathbf{3.659 \times 10^{-3}}$ | $1.768 \times 10^{-2}$ | $1.891 \times 10^{-2}$ |

## C.3 DETAILED TIME AND MEMORY COST COMPARSION

This section provides a detailed analysis of computational efficiency across baseline methods in terms of both runtime and memory consumption. The quantitative results are presented in Table 22.

During training, all models were evaluated using a standardized batch size of 16 to ensure fair comparison of GPU memory usage. Among the baselines, FNO and DeepONet exhibit relatively low memory consumption due to their ability to process full-field data ($128 \times 128$) with minimal intermediate state requirements. Although Physics-Informed Neural Networks (PINNs) possess a compact architecture ( 1M+ parameters), their point-wise computation approach results in memory usage comparable to the significantly larger FNO and DeepONet models ( 10M+ parameters), revealing their inherent inefficiency in memory utilization.

DiffusionPDE, the most parameter-heavy model ( 30M+), also demonstrates high memory usage, emphasizing the trade-off between model complexity and computational efficiency when addressing PDE-governed tasks.

For inference, we assess the per-sample latency across all baselines. DiffusionPDE shows significantly slower inference speeds due to its iterative denoising process (set to 10 steps in our implementation). Unlike the single-step prediction mechanism employed by other models, DiffusionPDE's sequential refinement framework inherently introduces higher computational overhead.

## C.4 DETAILED DATA SCALING RESULTS

The scaled experimental data for the Thermal-Fluid coupling are presented as Table 23.

## C.5 DETAILED TRICKS

**Trick 1: Quantile Normalization over Min-Max Normalization.** A prominent challenge in multi-physics datasets is the prevalence of long-tailed and asymmetric distributions, particularly in systems

Table 16: Baseline model performance on the Electro-Fluid task under different evaluation metrics.

| Metric | Fields | PINNs | DeepONet | FNO | DiffusionPDE |
|---|---|---|---|---|---|
| RMSE | $V$ | $4.897 \times 10^0$ | $\mathbf{5.325 \times 10^{-1}}$ | $6.125 \times 10^0$ | $9.311 \times 10^0$ |
| | $u$ | $6.561 \times 10^{-4}$ | $\mathbf{4.821 \times 10^{-4}}$ | $1.099 \times 10^{-3}$ | $1.789 \times 10^{-3}$ |
| | $v$ | $6.243 \times 10^{-4}$ | $\mathbf{3.069 \times 10^{-4}}$ | $6.266 \times 10^{-4}$ | $1.024 \times 10^{-3}$ |
| Relative $L_2$ Error | $V$ | $1.831 \times 10^{-1}$ | $\mathbf{1.962 \times 10^{-2}}$ | $2.368 \times 10^{-1}$ | $3.520 \times 10^{-1}$ |
| | $u$ | $1.160 \times 10^{-1}$ | $\mathbf{8.528 \times 10^{-2}}$ | $1.944 \times 10^{-1}$ | $3.164 \times 10^{-1}$ |
| | $v$ | $8.250 \times 10^{-1}$ | $\mathbf{4.054 \times 10^{-1}}$ | $8.279 \times 10^{-1}$ | $1.353 \times 10^0$ |
| MaxError | $V$ | $1.489 \times 10^1$ | $\mathbf{3.451 \times 10^0}$ | $1.451 \times 10^1$ | $2.250 \times 10^1$ |
| | $u$ | $\mathbf{5.788 \times 10^{-3}}$ | $6.408 \times 10^{-3}$ | $7.929 \times 10^{-3}$ | $1.002 \times 10^{-2}$ |
| | $v$ | $5.660 \times 10^{-3}$ | $\mathbf{3.629 \times 10^{-3}}$ | $5.568 \times 10^{-3}$ | $7.882 \times 10^{-3}$ |
| fRMSE low | $V$ | $4.451 \times 10^2$ | $\mathbf{3.439 \times 10^1}$ | $5.765 \times 10^2$ | $2.151 \times 10^4$ |
| | $u$ | $4.433 \times 10^{-2}$ | $\mathbf{5.354 \times 10^{-3}}$ | $6.897 \times 10^{-2}$ | $3.589 \times 10^0$ |
| | $v$ | $3.308 \times 10^{-2}$ | $\mathbf{3.667 \times 10^{-3}}$ | $3.186 \times 10^{-2}$ | $2.024 \times 10^0$ |
| fRMSE middle | $V$ | $4.125 \times 10^1$ | $\mathbf{1.322 \times 10^1}$ | $3.667 \times 10^1$ | $9.789 \times 10^2$ |
| | $u$ | $1.090 \times 10^{-2}$ | $\mathbf{1.087 \times 10^{-2}}$ | $1.788 \times 10^{-2}$ | $5.106 \times 10^{-1}$ |
| | $v$ | $9.664 \times 10^{-3}$ | $\mathbf{7.835 \times 10^{-3}}$ | $9.630 \times 10^{-3}$ | $2.961 \times 10^{-1}$ |
| fRMSE high | $V$ | $4.046 \times 10^1$ | $\mathbf{5.276 \times 10^0}$ | $4.710 \times 10^1$ | $9.462 \times 10^1$ |
| | $u$ | $6.834 \times 10^{-3}$ | $\mathbf{5.772 \times 10^{-3}}$ | $1.180 \times 10^{-2}$ | $6.638 \times 10^{-2}$ |
| | $v$ | $6.993 \times 10^{-3}$ | $\mathbf{3.662 \times 10^{-3}}$ | $7.076 \times 10^{-3}$ | $4.008 \times 10^{-2}$ |
| bRMSE | $V$ | $3.110 \times 10^0$ | $\mathbf{4.330 \times 10^{-1}}$ | $4.093 \times 10^0$ | $5.503 \times 10^0$ |
| | $u$ | $1.432 \times 10^{-4}$ | $\mathbf{7.184 \times 10^{-5}}$ | $1.365 \times 10^{-4}$ | $2.850 \times 10^{-4}$ |
| | $v$ | $3.704 \times 10^{-5}$ | $5.563 \times 10^{-5}$ | $\mathbf{8.273 \times 10^{-6}}$ | $5.800 \times 10^{-5}$ |

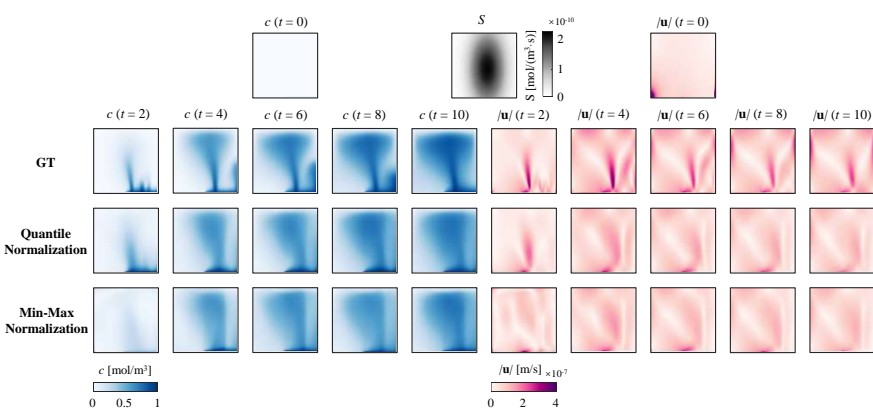

Figure 24: Quantile Normalization and Min-Max Normalization. Dataset: Mass Transfer-Fluid. Network: PINNs.

involving temporal dynamics such as Mass Transport–Fluid Coupling. Initial conditions generated via numerical solvers frequently contain singularities or outliers, which hinder conventional normalization methods.

To mitigate this issue, we replace the traditional min-max normalization with quantile normalization, which limits extreme values based on a specified percentile threshold (e.g., the 95th percentile). This approach effectively suppresses the influence of outliers and stabilizes the dynamic range of the input. Figure 24 illustrates the improvements gained through this method in the context of the Elder problem.

Table 17: Baseline model performance on the Magneto-Hydrodynamic task under different evaluation.

| Metric | Fields | PINNs | DeepONet | FNO | DiffusionPDE |
|---|---|---|---|---|---|
| RMSE | $J_x$ | $6.256 \times 10^{-1}$ | $2.653 \times 10^{0}$ | $\mathbf{2.987 \times 10^{-1}}$ | $4.051 \times 10^{-1}$ |
| | $J_y$ | $6.102 \times 10^{-1}$ | $1.419 \times 10^{0}$ | $\mathbf{3.932 \times 10^{-1}}$ | $5.204 \times 10^{-1}$ |
| | $J_z$ | $2.462 \times 10^{-1}$ | $7.486 \times 10^{-1}$ | $1.678 \times 10^{-1}$ | $\mathbf{1.468 \times 10^{-1}}$ |
| | $u$ | $1.096 \times 10^{-3}$ | $\mathbf{5.131 \times 10^{-5}}$ | $1.528 \times 10^{-3}$ | $1.471 \times 10^{-3}$ |
| | $v$ | $3.100 \times 10^{-4}$ | $\mathbf{2.504 \times 10^{-5}}$ | $3.140 \times 10^{-4}$ | $3.610 \times 10^{-4}$ |
| Relative $L_2$ Error | $J_x$ | $6.390 \times 10^{-2}$ | $2.709 \times 10^{-1}$ | $\mathbf{3.050 \times 10^{-2}}$ | $4.137 \times 10^{-2}$ |
| | $J_y$ | $4.065 \times 10^{-2}$ | $9.458 \times 10^{-2}$ | $\mathbf{2.619 \times 10^{-2}}$ | $3.467 \times 10^{-2}$ |
| | $J_z$ | $3.085 \times 10^{-1}$ | $9.382 \times 10^{-1}$ | $2.101 \times 10^{-1}$ | $\mathbf{1.836 \times 10^{-1}}$ |
| | $u$ | $4.080 \times 10^{0}$ | $\mathbf{1.044 \times 10^{-1}}$ | $2.127 \times 10^{0}$ | $4.920 \times 10^{0}$ |
| | $v$ | $2.029 \times 10^{0}$ | $\mathbf{1.460 \times 10^{-1}}$ | $1.319 \times 10^{0}$ | $4.296 \times 10^{0}$ |
| MaxError | $J_x$ | $1.452 \times 10^{1}$ | $9.271 \times 10^{1}$ | $\mathbf{1.187 \times 10^{1}}$ | $1.500 \times 10^{1}$ |
| | $J_y$ | $1.932 \times 10^{1}$ | $7.612 \times 10^{1}$ | $\mathbf{8.684 \times 10^{0}}$ | $1.110 \times 10^{1}$ |
| | $J_z$ | $5.910 \times 10^{0}$ | $4.142 \times 10^{1}$ | $5.878 \times 10^{0}$ | $\mathbf{5.584 \times 10^{0}}$ |
| | $u$ | $2.879 \times 10^{-3}$ | $\mathbf{3.922 \times 10^{-4}}$ | $3.354 \times 10^{-3}$ | $4.073 \times 10^{-3}$ |
| | $v$ | $1.059 \times 10^{-3}$ | $\mathbf{1.763 \times 10^{-4}}$ | $1.024 \times 10^{-3}$ | $1.312 \times 10^{-3}$ |
| fRMSE low | $J_x$ | $1.505 \times 10^{1}$ | $\mathbf{1.149 \times 10^{1}}$ | $1.458 \times 10^{1}$ | $5.023 \times 10^{2}$ |
| | $J_y$ | $2.096 \times 10^{1}$ | $\mathbf{1.199 \times 10^{1}}$ | $3.828 \times 10^{1}$ | $1.037 \times 10^{3}$ |
| | $J_z$ | $\mathbf{2.220 \times 10^{0}}$ | $3.416 \times 10^{0}$ | $1.125 \times 10^{1}$ | $4.605 \times 10^{1}$ |
| | $u$ | $1.054 \times 10^{-1}$ | $\mathbf{2.349 \times 10^{-3}}$ | $1.833 \times 10^{-1}$ | $3.431 \times 10^{0}$ |
| | $v$ | $3.037 \times 10^{-2}$ | $\mathbf{1.130 \times 10^{-3}}$ | $3.079 \times 10^{-2}$ | $8.219 \times 10^{-1}$ |
| fRMSE middle | $J_x$ | $1.336 \times 10^{1}$ | $3.107 \times 10^{1}$ | $\mathbf{4.524 \times 10^{0}}$ | $1.455 \times 10^{2}$ |
| | $J_y$ | $9.983 \times 10^{0}$ | $1.952 \times 10^{1}$ | $\mathbf{2.972 \times 10^{0}}$ | $8.263 \times 10^{1}$ |
| | $J_z$ | $2.509 \times 10^{0}$ | $6.396 \times 10^{0}$ | $\mathbf{1.802 \times 10^{0}}$ | $3.087 \times 10^{1}$ |
| | $u$ | $6.764 \times 10^{-3}$ | $1.406 \times 10^{-3}$ | $3.882 \times 10^{-3}$ | $7.255 \times 10^{-2}$ |
| | $v$ | $1.822 \times 10^{-3}$ | $\mathbf{7.745 \times 10^{-4}}$ | $1.758 \times 10^{-3}$ | $4.084 \times 10^{-2}$ |
| fRMSE high | $J_x$ | $7.380 \times 10^{0}$ | $3.293 \times 10^{1}$ | $\mathbf{3.491 \times 10^{0}}$ | $3.667 \times 10^{1}$ |
| | $J_y$ | $7.283 \times 10^{0}$ | $1.761 \times 10^{1}$ | $\mathbf{2.937 \times 10^{0}}$ | $3.103 \times 10^{1}$ |
| | $J_z$ | $2.880 \times 10^{0}$ | $9.307 \times 10^{0}$ | $\mathbf{1.569 \times 10^{0}}$ | $1.816 \times 10^{1}$ |
| | $u$ | $8.508 \times 10^{-3}$ | $\mathbf{5.792 \times 10^{-4}}$ | $9.616 \times 10^{-3}$ | $4.551 \times 10^{-3}$ |
| | $v$ | $3.585 \times 10^{-3}$ | $\mathbf{2.875 \times 10^{-4}}$ | $3.608 \times 10^{-3}$ | $2.236 \times 10^{-3}$ |
| bRMSE | $J_x$ | $1.187 \times 10^{0}$ | $1.130 \times 10^{1}$ | $\mathbf{2.948 \times 10^{-1}}$ | $6.760 \times 10^{-1}$ |
| | $J_y$ | $1.305 \times 10^{0}$ | $5.092 \times 10^{0}$ | $\mathbf{3.888 \times 10^{-1}}$ | $5.254 \times 10^{-1}$ |
| | $J_z$ | $4.381 \times 10^{-1}$ | $3.782 \times 10^{0}$ | $1.151 \times 10^{-1}$ | $\mathbf{1.267 \times 10^{-1}}$ |
| | $u$ | $7.070 \times 10^{-4}$ | $\mathbf{8.950 \times 10^{-5}}$ | $8.300 \times 10^{-4}$ | $7.040 \times 10^{-4}$ |
| | $v$ | $1.220 \times 10^{-4}$ | $\mathbf{3.943 \times 10^{-5}}$ | $1.230 \times 10^{-4}$ | $1.170 \times 10^{-4}$ |

**Trick 2: Auto-Balanced Weighting for Coupled PDE Residuals.** In multiphysics systems governed by multiple coupled PDEs, supervision signals often exhibit large discrepancies in magnitude, complicating the tuning of loss weights and frequently leading to suboptimal convergence. For instance, in the Thermo-Fluid Coupling scenario within DiffusionPDE, the temperature and velocity fields may differ in scale by several orders of magnitude (e.g., up to a ratio of $1 : 10^{-5}$). This imbalance results in gradient magnitudes that differ substantially across fields, causing fixed-weight loss formulations to hinder performance.

To address this challenge, we propose an automatic balancing mechanism that scales PDE residuals according to their respective $L_2$ norms, bringing them into comparable numerical ranges. This dynamic weighting strategy eliminates the need for manual tuning and enables simultaneous optimization across all PDE components. Empirical results, as visualized in Figure 25, confirm the effectiveness of this approach, particularly in improving model robustness and prediction accuracy in DiffusionPDE.

Table 18: Baseline model performance on the Acoustic–Structure task under different evaluation.

| Metric | Fields | PINNs | DeepONet | FNO | DiffusionPDE |
|---|---|---|---|---|---|
| RMSE (Re/Im) | $p$ | $(5.7/5.7) \times 10^{-1}$ | $(\mathbf{6.9}/\mathbf{6.6}) \times 10^{-2}$ | $(6.2/5.8) \times 10^{-1}$ | $(4.5/5.4) \times 10^{-1}$ |
| | $S_{xx}$ | $(1.7/1.7) \times 10^{-1}$ | $(\mathbf{4.6}/\mathbf{3.6}) \times 10^{-2}$ | $(1.6/1.8) \times 10^{-1}$ | $(4.9/5.1) \times 10^{-1}$ |
| | $S_{xy}$ | $(3.6/4.3) \times 10^{-2}$ | $(\mathbf{7.4}/\mathbf{7.3}) \times 10^{-3}$ | $(4.0/4.1) \times 10^{-2}$ | $(5.9/6.9) \times 10^{-2}$ |
| | $S_{yy}$ | $(1.7/1.9) \times 10^{-1}$ | $(\mathbf{4.6}/\mathbf{3.6}) \times 10^{-2}$ | $(1.7/1.9) \times 10^{-1}$ | $(6.9/5.1) \times 10^{-1}$ |
| | $x$ | $(3.5/3.1) \times 10^{2}$ | $(\mathbf{1.9}/\mathbf{3.1}) \times 10^{0}$ | $(3.7/3.4) \times 10^{2}$ | $(3.2/3.5) \times 10^{2}$ |
| | $y$ | $(2.5/2.2) \times 10^{2}$ | $(\mathbf{1.7}/\mathbf{2.3}) \times 10^{0}$ | $(2.6/2.4) \times 10^{2}$ | $(2.2/3.4) \times 10^{2}$ |
| Relative $L_2$ Error (Re/Im) | $p$ | $(5.0/4.4) \times 10^{-1}$ | $(\mathbf{6.2}/\mathbf{5.3}) \times 10^{-2}$ | $(5.0/4.0) \times 10^{-1}$ | $(4.7/5.0) \times 10^{-1}$ |
| | $S_{xx}$ | $(6.6/6.0) \times 10^{-1}$ | $(\mathbf{2.0}/\mathbf{1.1}) \times 10^{-1}$ | $(5.2/4.7) \times 10^{-1}$ | $(2.5/1.8) \times 10^{0}$ |
| | $S_{xy}$ | $(5.7/1.7) \times 10^{-1}$ | $(\mathbf{9.0}/\mathbf{2.6}) \times 10^{-2}$ | $(5.3/7.6) \times 10^{-1}$ | $(8.1/3.4) \times 10^{-1}$ |
| | $S_{yy}$ | $(5.7/6.5) \times 10^{-1}$ | $(\mathbf{1.6}/\mathbf{1.1}) \times 10^{-1}$ | $(5.1/4.9) \times 10^{-1}$ | $(4.2/1.8) \times 10^{0}$ |
| | $x$ | $(4.1/1.4) \times 10^{0}$ | $(\mathbf{1.5}/6.7) \times 10^{-1}$ | $(9.2/\mathbf{1.2}) \times 10^{-1}$ | $(4.3/1.4) \times 10^{0}$ |
| | $y$ | $(3.3/9.4) \times 10^{0}$ | $(\mathbf{1.7}/\mathbf{7.6}) \times 10^{-1}$ | $(8.5/7.8) \times 10^{-1}$ | $(3.4/1.4) \times 10^{0}$ |
| MaxError (Re/Im) | $p$ | $(1.9/2.0) \times 10^{0}$ | $(\mathbf{8.7}/\mathbf{7.3}) \times 10^{-1}$ | $(1.8/1.7) \times 10^{0}$ | $(7.1/8.0) \times 10^{0}$ |
| | $S_{xx}$ | $(1.7/1.8) \times 10^{0}$ | $(\mathbf{8.9}/\mathbf{6.8}) \times 10^{-1}$ | $(1.6/1.6) \times 10^{0}$ | $(4.2/4.7) \times 10^{0}$ |
| | $S_{xy}$ | $(3.2/3.7) \times 10^{-1}$ | $(\mathbf{8.5}/\mathbf{8.9}) \times 10^{-2}$ | $(3.4/3.4) \times 10^{-1}$ | $(4.4/5.0) \times 10^{-1}$ |
| | $S_{yy}$ | $(1.7/1.8) \times 10^{0}$ | $(\mathbf{8.8}/\mathbf{6.9}) \times 10^{-1}$ | $(1.6/1.6) \times 10^{0}$ | $(8.0/4.6) \times 10^{0}$ |
| | $x$ | $(2.2/1.9) \times 10^{2}$ | $(6.0/\mathbf{1.4}) \times 10^{2}$ | $(\mathbf{1.7}/1.6) \times 10^{2}$ | $(1.7/1.8) \times 10^{2}$ |
| | $y$ | $(1.5/1.5) \times 10^{2}$ | $(\mathbf{4.8}/\mathbf{8.5}) \times 10^{1}$ | $(1.3/1.2) \times 10^{2}$ | $(1.3/2.1) \times 10^{2}$ |
| fRMSE low (Re/Im) | $p$ | $(\mathbf{6.8}/\mathbf{7.5}) \times 10^{0}$ | $(8.7/8.3) \times 10^{0}$ | $(5.0/5.1) \times 10^{1}$ | $(4.4/5.5) \times 10^{2}$ |
| | $S_{xx}$ | $(\mathbf{1.5}/\mathbf{2.0}) \times 10^{0}$ | $(1.9/2.0) \times 10^{0}$ | $(9.5/12) \times 10^{0}$ | $(2.3/2.4) \times 10^{2}$ |
| | $S_{xy}$ | $(\mathbf{3.2}/\mathbf{3.7}) \times 10^{-1}$ | $(3.7/4.5) \times 10^{-1}$ | $(2.7/2.9) \times 10^{0}$ | $(5.0/6.8) \times 10^{1}$ |
| | $S_{yy}$ | $(\mathbf{1.7}/2.6) \times 10^{0}$ | $(1.9/\mathbf{1.9}) \times 10^{0}$ | $(1.0/1.4) \times 10^{1}$ | $(2.4/3.0) \times 10^{2}$ |
| | $x$ | $(4.9/2.4) \times 10^{2}$ | $(\mathbf{2.1}/\mathbf{1.9}) \times 10^{2}$ | $(2.9/2.5) \times 10^{3}$ | $(4.6/4.0) \times 10^{4}$ |
| | $y$ | $(2.4/2.9) \times 10^{2}$ | $(\mathbf{1.5}/\mathbf{1.9}) \times 10^{2}$ | $(1.9/1.8) \times 10^{3}$ | $(2.6/3.2) \times 10^{4}$ |
| fRMSE middle (Re/Im) | $p$ | $(\mathbf{6.2}/\mathbf{7.1}) \times 10^{-1}$ | $(3.4/3.4) \times 10^{0}$ | $(4.5/4.5) \times 10^{0}$ | $(7.6/7.5) \times 10^{1}$ |
| | $S_{xx}$ | $(\mathbf{4.7}/\mathbf{5.8}) \times 10^{-1}$ | $(2.4/2.1) \times 10^{0}$ | $(2.8/3.4) \times 10^{0}$ | $(8.2/9.0) \times 10^{1}$ |
| | $S_{xy}$ | $(\mathbf{8.1}/\mathbf{6.8}) \times 10^{-2}$ | $(5.0/5.8) \times 10^{-1}$ | $(4.0/4.1) \times 10^{-1}$ | $(1.6/1.3) \times 10^{1}$ |
| | $S_{yy}$ | $(\mathbf{4.8}/\mathbf{5.9}) \times 10^{-1}$ | $(2.3/2.1) \times 10^{0}$ | $(2.8/3.4) \times 10^{0}$ | $(1.1/9.1) \times 10^{2}$ |
| | $x$ | $(\mathbf{9.0}/\mathbf{4.7}) \times 10^{1}$ | $(9.3/9.1) \times 10^{1}$ | $(5.3/4.4) \times 10^{2}$ | $(8.8/6.7) \times 10^{3}$ |
| | $y$ | $(\mathbf{5.2}/\mathbf{5.9}) \times 10^{1}$ | $(8.2/9.3) \times 10^{1}$ | $(3.5/3.3) \times 10^{2}$ | $(5.5/8.0) \times 10^{3}$ |
| fRMSE high (Re/Im) | $p$ | $(\mathbf{8.2}/\mathbf{8.7}) \times 10^{-1}$ | $(2.0/1.8) \times 10^{0}$ | $(6.3/5.7) \times 10^{0}$ | $(4.5/5.2) \times 10^{1}$ |
| | $S_{xx}$ | $(\mathbf{3.0}/\mathbf{3.4}) \times 10^{-1}$ | $(1.4/1.1) \times 10^{0}$ | $(1.9/2.0) \times 10^{0}$ | $(6.0/6.1) \times 10^{1}$ |
| | $S_{xy}$ | $(\mathbf{7.0}/\mathbf{5.1}) \times 10^{-2}$ | $(2.1/2.0) \times 10^{-1}$ | $(3.7/3.9) \times 10^{-1}$ | $(6.4/7.5) \times 10^{0}$ |
| | $S_{yy}$ | $(\mathbf{3.2}/\mathbf{3.9}) \times 10^{-1}$ | $(1.4/1.1) \times 10^{0}$ | $(2.0/2.2) \times 10^{0}$ | $(1.4/6.0) \times 10^{2}$ |
| | $x$ | $(6.9/\mathbf{3.5}) \times 10^{1}$ | $(\mathbf{5.9}/9.4) \times 10^{1}$ | $(4.0/3.3) \times 10^{2}$ | $(2.7/3.5) \times 10^{3}$ |
| | $y$ | $(\mathbf{3.5}/\mathbf{4.0}) \times 10^{1}$ | $(5.0/6.5) \times 10^{1}$ | $(2.6/2.5) \times 10^{2}$ | $(2.0/3.6) \times 10^{3}$ |
| bRMSE (Re/Im) | $p$ | $(2.6/2.5) \times 10^{-1}$ | $(\mathbf{7.3}/\mathbf{8.3}) \times 10^{-2}$ | $(2.8/2.9) \times 10^{-1}$ | $(4.5/5.1) \times 10^{-1}$ |
| | $S_{xx}$ | $(1.1/1.1) \times 10^{-2}$ | $(2.5/1.8) \times 10^{-2}$ | $(\mathbf{1.9}/\mathbf{5.2}) \times 10^{-3}$ | $(3.2/2.6) \times 10^{-1}$ |
| | $S_{xy}$ | $(2.2/3.3) \times 10^{-3}$ | $(4.7/3.7) \times 10^{-3}$ | $(\mathbf{7.2}/\mathbf{4.5}) \times 10^{-4}$ | $(3.9/5.0) \times 10^{-2}$ |
| | $S_{yy}$ | $(9.7/\mathbf{1.2}) \times 10^{-3}$ | $(2.5/1.8) \times 10^{-2}$ | $(\mathbf{1.9}/3.0) \times 10^{-3}$ | $(6.8/3.1) \times 10^{-1}$ |
| | $x$ | $(1.2/1.5) \times 10^{0}$ | $(8.4/\mathbf{1.0}) \times 10^{-1}$ | $(\mathbf{3.8}/7.3) \times 10^{-1}$ | $(2.1/2.7) \times 10^{1}$ |
| | $y$ | $(8.0/1.1) \times 10^{-1}$ | $(6.9/8.3) \times 10^{-1}$ | $(\mathbf{8.2}/\mathbf{5.5}) \times 10^{-2}$ | $(1.6/2.6) \times 10^{1}$ |

Table 19: Baseline model performance on the Mass Transport–Fluid Coupling task under different evaluation metrics.

| Metric | Fields | PINNs | DeepONet | FNO | DiffusionPDE |
|---|---|---|---|---|---|
| RMSE | $u$ | $3.506 \times 10^{-8}$ | $3.332 \times 10^{-8}$ | $\mathbf{2.878 \times 10^{-8}}$ | $1.760 \times 10^{-7}$ |
| | $v$ | $5.486 \times 10^{-8}$ | $5.479 \times 10^{-8}$ | $\mathbf{4.840 \times 10^{-8}}$ | $4.333 \times 10^{-7}$ |
| | $c$ | $\mathbf{8.355 \times 10^{-2}}$ | $8.795 \times 10^{-2}$ | $8.969 \times 10^{-2}$ | $2.986 \times 10^{-1}$ |
| Relative $L_2$ Error | $u$ | $\mathbf{2.055 \times 10^{-1}}$ | $5.439 \times 10^{-1}$ | $4.645 \times 10^{-1}$ | $2.758 \times 10^{0}$ |
| | $v$ | $\mathbf{1.376 \times 10^{-1}}$ | $8.182 \times 10^{-1}$ | $7.244 \times 10^{-1}$ | $6.298 \times 10^{0}$ |
| | $c$ | $2.713 \times 10^{-1}$ | $\mathbf{2.617 \times 10^{-1}}$ | $2.623 \times 10^{-1}$ | $8.340 \times 10^{-1}$ |
| MaxError | $u$ | $2.118 \times 10^{-7}$ | $3.471 \times 10^{-7}$ | $\mathbf{1.668 \times 10^{-7}}$ | $4.929 \times 10^{-7}$ |
| | $v$ | $2.428 \times 10^{-7}$ | $2.448 \times 10^{-7}$ | $\mathbf{2.108 \times 10^{-7}}$ | $7.708 \times 10^{-7}$ |
| | $c$ | $3.840 \times 10^{-1}$ | $4.153 \times 10^{-1}$ | $\mathbf{3.498 \times 10^{-1}}$ | $1.022 \times 10^{0}$ |
| fRMSE low | $u$ | $4.533 \times 10^{-5}$ | $\mathbf{4.107 \times 10^{-5}}$ | $4.838 \times 10^{-5}$ | $4.033 \times 10^{-4}$ |
| | $v$ | $9.621 \times 10^{-5}$ | $9.398 \times 10^{-5}$ | $\mathbf{9.141 \times 10^{-5}}$ | $1.100 \times 10^{-3}$ |
| | $c$ | $1.766 \times 10^{2}$ | $\mathbf{1.575 \times 10^{2}}$ | $2.005 \times 10^{2}$ | $6.416 \times 10^{2}$ |
| fRMSE middle | $u$ | $1.434 \times 10^{-5}$ | $1.424 \times 10^{-5}$ | $\mathbf{1.069 \times 10^{-5}}$ | $2.150 \times 10^{-5}$ |
| | $v$ | $3.033 \times 10^{-5}$ | $3.142 \times 10^{-5}$ | $\mathbf{2.627 \times 10^{-5}}$ | $2.780 \times 10^{-5}$ |
| | $c$ | $4.004 \times 10^{1}$ | $4.125 \times 10^{1}$ | $\mathbf{3.407 \times 10^{1}}$ | $7.365 \times 10^{1}$ |
| fRMSE high | $u$ | $9.706 \times 10^{-6}$ | $9.289 \times 10^{-6}$ | $7.346 \times 10^{-6}$ | $\mathbf{2.140 \times 10^{-6}}$ |
| | $v$ | $1.355 \times 10^{-5}$ | $1.362 \times 10^{-5}$ | $1.150 \times 10^{-5}$ | $\mathbf{1.998 \times 10^{-6}}$ |
| | $c$ | $2.285 \times 10^{1}$ | $2.193 \times 10^{1}$ | $1.903 \times 10^{1}$ | $\mathbf{7.875 \times 10^{0}}$ |
| bRMSE | $u$ | $4.963 \times 10^{-8}$ | $5.309 \times 10^{-8}$ | $\mathbf{3.717 \times 10^{-8}}$ | $1.850 \times 10^{-7}$ |
| | $v$ | $6.766 \times 10^{-8}$ | $7.223 \times 10^{-8}$ | $\mathbf{5.969 \times 10^{-8}}$ | $4.166 \times 10^{-7}$ |
| | $c$ | $\mathbf{9.215 \times 10^{-2}}$ | $1.026 \times 10^{-1}$ | $9.290 \times 10^{-2}$ | $4.245 \times 10^{-1}$ |

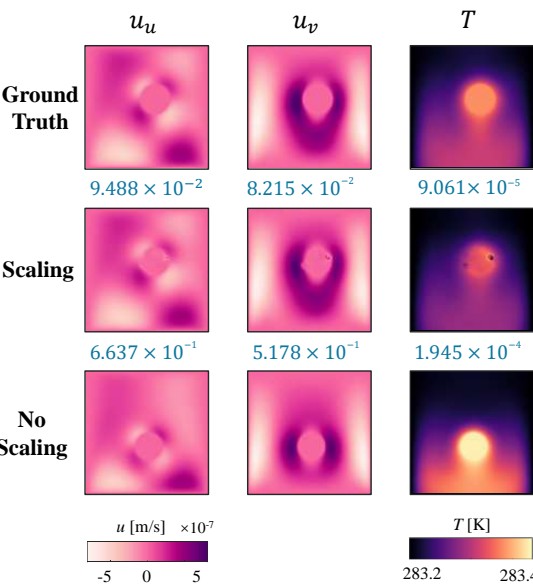

Figure 25: Auto-weighted learning across coupled PDE components boosts model accuracy. Dataset: Electro-Thermal. Network: DiffusionPDE.

Table 20: Performance of baseline models under noisy and noise-free conditions. Metric: Relative $L_2$ Error. Dataset: Electro–Thermal.

| Conditions | Fields | PINNs | DeepONet | FNO | DiffusionPDE |
|---|---|---|---|---|---|
| Noise-Free | $\text{Re}(E_z)$ | $4.387 \times 10^{-1}$ | $2.573 \times 10^{-1}$ | $4.388 \times 10^{-1}$ | $5.499 \times 10^{-1}$ |
| | $\text{Im}(E_z)$ | $4.469 \times 10^{-1}$ | $2.617 \times 10^{-1}$ | $4.474 \times 10^{-1}$ | $5.330 \times 10^{-1}$ |
| | $T$ | $1.573 \times 10^{-3}$ | $2.317 \times 10^{-3}$ | $1.482 \times 10^{-3}$ | $1.560 \times 10^{-3}$ |
| Gaussian Noise $\mu = 0.1$, $\sigma = 0.1$ | $\text{Re}(E_z)$ | $4.406 \times 10^{-1}$ | $4.043 \times 10^{-1}$ | $4.388 \times 10^{-1}$ | $5.740 \times 10^{-1}$ |
| | $\text{Im}(E_z)$ | $4.485 \times 10^{-1}$ | $4.216 \times 10^{-1}$ | $4.475 \times 10^{-1}$ | $5.480 \times 10^{-1}$ |
| | $T$ | $1.563 \times 10^{-3}$ | $2.614 \times 10^{-3}$ | $1.483 \times 10^{-3}$ | $1.601 \times 10^{-3}$ |
| Gaussian Noise $\mu = 0.1$, $\sigma = 0.2$ | $\text{Re}(E_z)$ | $4.449 \times 10^{-1}$ | $4.869 \times 10^{-1}$ | $4.388 \times 10^{-1}$ | $5.814 \times 10^{-1}$ |
| | $\text{Im}(E_z)$ | $4.513 \times 10^{-1}$ | $5.058 \times 10^{-1}$ | $4.475 \times 10^{-1}$ | $5.535 \times 10^{-1}$ |
| | $T$ | $1.553 \times 10^{-3}$ | $2.919 \times 10^{-3}$ | $1.483 \times 10^{-3}$ | $1.607 \times 10^{-3}$ |
| Gaussian Noise $\mu = 0.1$, $\sigma = 0.3$ | $\text{Re}(E_z)$ | $4.543 \times 10^{-1}$ | $6.326 \times 10^{-1}$ | $4.389 \times 10^{-1}$ | $5.835 \times 10^{-1}$ |
| | $\text{Im}(E_z)$ | $4.566 \times 10^{-1}$ | $6.526 \times 10^{-1}$ | $4.475 \times 10^{-1}$ | $5.565 \times 10^{-1}$ |
| | $T$ | $1.548 \times 10^{-3}$ | $3.370 \times 10^{-3}$ | $1.483 \times 10^{-3}$ | $1.612 \times 10^{-3}$ |
| Poison Noise $\mu = 0.1$, $\sigma = 0.1$ | $\text{Re}(E_z)$ | $4.451 \times 10^{-1}$ | $8.848 \times 10^{-1}$ | $4.388 \times 10^{-1}$ | $5.679 \times 10^{-1}$ |
| | $\text{Im}(E_z)$ | $4.501 \times 10^{-1}$ | $8.848 \times 10^{-1}$ | $4.475 \times 10^{-1}$ | $5.461 \times 10^{-1}$ |
| | $T$ | $1.565 \times 10^{-3}$ | $4.903 \times 10^{-3}$ | $1.483 \times 10^{-3}$ | $1.618 \times 10^{-3}$ |
| Poison Noise $\mu = 0.1$, $\sigma = 0.2$ | $\text{Re}(E_z)$ | $4.489 \times 10^{-1}$ | $9.584 \times 10^{-1}$ | $4.388 \times 10^{-1}$ | $5.779 \times 10^{-1}$ |
| | $\text{Im}(E_z)$ | $4.528 \times 10^{-1}$ | $1.030 \times 10^{0}$ | $4.475 \times 10^{-1}$ | $5.547 \times 10^{-1}$ |
| | $T$ | $1.557 \times 10^{-3}$ | $5.030 \times 10^{-3}$ | $1.483 \times 10^{-3}$ | $1.618 \times 10^{-3}$ |
| Poison Noise $\mu = 0.1$, $\sigma = 0.3$ | $\text{Re}(E_z)$ | $4.571 \times 10^{-1}$ | $1.017 \times 10^{0}$ | $4.389 \times 10^{-1}$ | $5.809 \times 10^{-1}$ |
| | $\text{Im}(E_z)$ | $4.580 \times 10^{-1}$ | $1.096 \times 10^{0}$ | $4.475 \times 10^{-1}$ | $5.577 \times 10^{-1}$ |
| | $T$ | $1.555 \times 10^{-3}$ | $5.064 \times 10^{-3}$ | $1.483 \times 10^{-3}$ | $1.618 \times 10^{-3}$ |

Table 21: Multiphysics data incorporates more comprehensive physical priors and demonstrates superior performance. Metric: Relative $L_2$ Error. Dataset: Electro-Thermal.

| | PINNs | DeepONet | FNO | DiffusionPDE |
|---|---|---|---|---|
| $\text{Re}/\text{Im}(E_z)$ (Inc) | $(4.38/4.47) \times 10^{-1}$ | $(4.42/4.51) \times 10^{-1}$ | $(4.39/4.48) \times 10^{-1}$ | $(5.16/5.25) \times 10^{-1}$ |
| $\text{Re}/\text{Im}(E_z)$ (C) | $(4.39/4.47) \times 10^{-1}$ | $(2.57/2.62) \times 10^{-1}$ | $(4.39/4.48) \times 10^{-1}$ | $(5.50/5.33) \times 10^{-1}$ |

Table 22: Computational efficiency comparison of different methods on the Electro-Thermal problem.

| Method | Memory (MiB) | Inference Time (s) |
|---|---|---|
| PINNs | 11,356 | 0.2022 |
| FNO | 11,160 | 0.1722 |
| DeepONet | 10,572 | 0.4927 |
| DiffusionPDE | 24,104 | 12.9 |

Table 23: Relative $L_2$ error performance of baseline models across varying data scales. Metric: Relative $L_2$ Error. Dataset: Thermal-Fluid. D-ONet denotes DeepONet, Diff-PDE refers to DiffusionPDE.

| RE | Fields | 300 | 1,000 | 3,000 | 10,000 | 30,000 |
|---|---|---|---|---|---|---|
| PINNs | $u$ | $9.41 \times 10^{-1}$ | $8.56 \times 10^{-1}$ | $7.84 \times 10^{-1}$ | $6.20 \times 10^{-1}$ | $\mathbf{4.78 \times 10^{-1}}$ |
| | $v$ | $1.01 \times 10^{0}$ | $9.54 \times 10^{-1}$ | $6.98 \times 10^{-1}$ | $5.00 \times 10^{-1}$ | $\mathbf{3.66 \times 10^{-1}}$ |
| | $T$ | $2.66 \times 10^{-4}$ | $2.05 \times 10^{-4}$ | $1.60 \times 10^{-4}$ | $1.33 \times 10^{-4}$ | $\mathbf{1.20 \times 10^{-4}}$ |
| D-ONet | $u$ | $4.49 \times 10^{-1}$ | $3.08 \times 10^{-1}$ | $1.98 \times 10^{-1}$ | $1.11 \times 10^{-1}$ | $\mathbf{7.57 \times 10^{-2}}$ |
| | $v$ | $4.08 \times 10^{-1}$ | $2.90 \times 10^{-1}$ | $1.83 \times 10^{-1}$ | $9.70 \times 10^{-2}$ | $\mathbf{7.13 \times 10^{-2}}$ |
| | $T$ | $1.03 \times 10^{-4}$ | $5.24 \times 10^{-5}$ | $3.58 \times 10^{-5}$ | $1.78 \times 10^{-5}$ | $\mathbf{1.20 \times 10^{-5}}$ |
| FNO | $u$ | $5.04 \times 10^{-1}$ | $\mathbf{4.56 \times 10^{-1}}$ | $4.57 \times 10^{-1}$ | $4.57 \times 10^{-1}$ | $4.57 \times 10^{-1}$ |
| | $v$ | $6.91 \times 10^{-1}$ | $4.07 \times 10^{-1}$ | $4.06 \times 10^{-1}$ | $4.05 \times 10^{-1}$ | $\mathbf{4.05 \times 10^{-1}}$ |
| | $T$ | $2.39 \times 10^{-4}$ | $1.41 \times 10^{-4}$ | $1.22 \times 10^{-4}$ | $1.42 \times 10^{-4}$ | $\mathbf{1.42 \times 10^{-4}}$ |
| Diff-PDE | $u$ | $1.36 \times 10^{0}$ | $5.28 \times 10^{-1}$ | $2.51 \times 10^{-1}$ | $1.98 \times 10^{-1}$ | $\mathbf{1.73 \times 10^{-1}}$ |
| | $v$ | $2.71 \times 10^{-1}$ | $2.09 \times 10^{-1}$ | $2.35 \times 10^{-1}$ | $\mathbf{1.87 \times 10^{-1}}$ | $2.01 \times 10^{-1}$ |
| | $T$ | $1.40 \times 10^{-4}$ | $1.09 \times 10^{-4}$ | $1.14 \times 10^{-4}$ | $1.07 \times 10^{-4}$ | $\mathbf{6.20 \times 10^{-5}}$ |

