# OpenReview forum: "Multiphysics Bench: Benchmarking and Investigating Scientific Machine Learning for Multiphysics PDEs"
_ICLR.cc/2026/Conference — Submitted to ICLR 2026_

### Official Review · Reviewer_rneb · 2025-10-27

**Soundness:** 3
**Presentation:** 3
**Contribution:** 2
**Rating:** 6
**Confidence:** 2

**Summary:**

This paper introduces the first general multiphysics dataset, the Multiphysics Bench, for PDE solving. It includes six canonical coupled systems (e.g., electro-thermal, thermo-fluid, magneto-hydrodynamic, etc.) generated via finite element simulations. The authors evaluate four representative neural PDE solvers, PINNs, FNO, DeepONet, and DiffusionPDE, on the Multiphysics Bench.

**Strengths:**

1. The paper is well-written, clearly motivated, and easy to follow.
2. Multiphysics problems are largely underexplored in SciML benchmarks. While many benchmarks exist for single-physics problems, multiphysics systems are far more representative of real-world challenges but have lacked a standardized dataset for evaluation.
3. The experiments are systematic and provide valuable insights. The failure of general-purpose PDE solvers like FNO stresses the importance of investigating into new PDE solvers for multi-physics PDEs.

**Weaknesses:**

1. Since the Multiphysics Bench's focus is on coupled PDEs, the evaluation would be significantly stronger if it included at least one baseline specifically designed for such coupled systems. One possible solver might be [1].
2. The paper's primary contribution appears to be the introduction of new datasets. The work would be substantially strengthened by the inclusion of a novel PDE solver tailored to the specific challenges these datasets present.
3. The citation style is incorrect for ICLR guidelines (\citep should be used for parenthetical citations).

[1] Xiao, X., Cao, D., Yang, R., Gupta, G., Liu, G., Yin, C., ... & Bogdan, P. (2023). Coupled multiwavelet neural operator learning for coupled partial differential equations. arXiv preprint arXiv:2303.02304.

**Questions:**

1. What do you think is the difference between coupled PDE in a single-physics domain compared to multi-physics one?
2. The authors conduct experiments on the 'Complete vs. Incomplete Physical Priors' section to prove the significance of understanding coupling PDEs instead of focusing on the sliced PDEs. However, the distinction between complete and incomplete prior isn't that obvious. Could authors provide stronger proof about the importance of physcial priors here? And also, could the authors provide an explanation on why DeepONet works better with incompete physical prior.

---

> ### Author Response · Authors · 2025-11-23
> **Response to Reviewer rneb – Part 1**
>
> We sincerely appreciate the reviewer’s positive evaluation of our work. Below, we provide detailed explanations to address your concerns.
> ***
> **W1:** Since the Multiphysics Bench's focus is on coupled PDEs, the evaluation would be significantly stronger if it included at least one baseline specifically designed for such coupled systems. One possible solver might be [1].
>
> **A1:** We recently tested **M2PDE** [2], which is a novel diffusion-based framework designed to learn the conditional distributions of diverse physical processes from decoupled datasets. During the inference phase, M2PDE integrates these learned conditional distributions through iterative optimization to generate globally consistent solutions that respect the underlying physical coupling relationships. As a case study, we apply M2PDE to an electro-thermal coupling problem, where it is used to simultaneously predict both the electric field and the temperature field. Since the input–output structure of this model differs from that used in our baseline experiments, the results are reported separately in the table below. The mean squared errors （corresponding relative errors) on the test dataset are summarized as follows:
> |        | **Re($E_z$)**        | **Im($E_z$)**        |   **$T$**   |
> |--------|-----------|-----------|-----------|
> | **T2E**    | 1.483×10⁰ | 1.469×10⁰ | -         |
> | **E2T**   | -         | -         | 8.126×10⁻³ |
> | **M2PDE**  | 1.007×10⁰ | 1.011×10⁰ | 8.392×10⁻³ |
>
> Due to time constraints, we adopted a relatively small number of training steps, which may have limited the overall performance. Nevertheless, the results clearly show that even under limited training, M2PDE—with its iterative coupling across physical fields (M2PDE)—outperforms models based on decoupled field learning (T2E and E2T). This highlights the necessity of designing specialized methods tailored for coupled multiphysics problems. We will continue to refine this part of the evaluation, fully considering the methods in [1] and [2], and incorporate the results into the manuscript.
> ***
> **W2:** The paper's primary contribution appears to be the introduction of new datasets. The work would be substantially strengthened by the inclusion of a novel PDE solver tailored to the specific challenges these datasets present.
>
> **A2:** Thank you for your suggestion. The primary focus of this work is to establish a reliable benchmark for multiphysics PDE problems and to conduct a baseline investigation using several representative SciML methods. During this process, we have also identified and introduced several practical techniques for applying SciML models to multiphysics scenarios.
> In future work, we plan to develop a dedicated PDE solver specifically designed for the challenges presented by multiphysics problems, with the goal of further advancing the SciML community.
> ***
> **W3:** The citation style is incorrect for ICLR guidelines (\citep should be used for parenthetical citations).
>
> **A3:** We appreciate your comment and have updated the citation format accordingly in the revised manuscript.
> ***
> [1] Xiao, X., Cao, D., Yang, R., Gupta, G., Liu, G., Yin, C., ... & Bogdan, P. (2023). Coupled multiwavelet neural operator learning for coupled partial differential equations. arXiv preprint arXiv:2303.02304.
>
> [2] T. Zhang, Z. liu, et al., "M2PDE: Compositional Generative Multiphysics and Multi-component PDE Simulation," in The Forty-Second International Conference on Machine Learning (ICML) 2025.

---

> ### Author Response · Authors · 2025-11-23
> **Response to Reviewer rneb – Part 2**
>
> **Q1:** What do you think is the difference between coupled PDE in a single-physics domain compared to multi-physics one?
>
> **A4:** In single-physics domains, coupled PDEs typically describe interactions among different components of the same physical quantity—for example, velocity–pressure coupling in fluid dynamics or displacement–stress coupling in elasticity. Although the equations are coupled, they share the same underlying physical mechanism, time scales, and material properties, and their coupling structure is generally consistent. Therefore, the primary sources of difficulty lie in the degree of nonlinearity of the equations or the complexity of boundary conditions.
> In contrast, multi-physics coupling involves fundamentally different physical processes (e.g., electromagnetic–thermal–mechanical). Each domain follows distinct conservation laws and involves physical quantities with different dimensions (scalars, vectors, tensors), as well as entirely different time scales, spatial scales, and material dependencies. Energy transfer between fields, parameter dependencies (e.g., temperature-dependent conductivity), and cross-domain feedback create a strongly coupled system in which the PDEs are highly nonlinear, multiscale, and no longer uniformly separable. Consequently, solving coupled PDEs in multi-physics scenarios is far more challenging than in single-physics domains, for both traditional numerical methods and SciML-based approaches.
> We have incorporated this discussion into the **Multiphysics Problems** section of the manuscript.
> ***
> **Q2:** The authors conduct experiments on the 'Complete vs. Incomplete Physical Priors' section to prove the significance of understanding coupling PDEs instead of focusing on the sliced PDEs. However, the distinction between complete and incomplete prior isn't that obvious. Could authors provide stronger proof about the importance of physcial priors here? And also, could the authors provide an explanation on why DeepONet works better with incompete physical prior.
>
> **A5:** In experiment T (incomplete), DeepONet performs slightly better than in experiment T (complete). At first glance, this result appears counterintuitive, as incomplete data lack sufficient information to fully describe the temperature field T, especially in capturing perturbations arising from multiphysics coupling. In contrast, in Experiment T (complete), which contains full field information, one would theoretically expect higher prediction accuracy.
> This expectation is indeed confirmed by the results of the other three baseline models, though not always pronounced. However, DeepONet exhibits the opposite trend. We interpret this difference as highlighting a fundamental limitation of the current architecture in capturing multiphysics coupling effects. DeepONet is essentially an operator regression model: its strength lies in fitting a function family rather than explicitly modeling the underlying PDE structure. When the input is incomplete, the imposed physical constraints are reduced, allowing the network to rely more heavily on the statistical distribution of the training data to approximate the output. This can superficially appear as improved performance, but it does not indicate that the true coupled physics are captured; rather, the weaker physical constraints produce a smoother loss landscape, which is easier to optimize.
> A similar issue arises in other methods. Under the complete prior setting, the network must satisfy multiple physical terms simultaneously, including nonlinear terms, complex couplings, and gradients at different scales. This leads to gradients of varying magnitudes, competition among coupled terms, and optimization difficulties. In fact, the complete prior can make optimization harder. This phenomenon reflects the stiffness of multiphysics-coupled PDEs and the associated gradient instability. Fully exposing the physical prior reveals the network's inability to adapt to multiphysics problems, highlighting a current limitation of SciML methods in solving coupled PDEs.
> ***
> We hope that these additional experiments and explanations resolve the reviewer's concerns and questions. We are more than happy to discuss if the reviewer has further questions or comments.

---

> > ### Comment · Reviewer_rneb · 2025-11-25
> >
> > Thanks for addressing my concerns. I will maintain my score.

---

> > > ### Author Response · Authors · 2025-11-26
> > >
> > > Thank you for the reviewer’s response. We sincerely welcome any further questions or discussions!

---

### Official Review · Reviewer_6qbz · 2025-10-31

**Soundness:** 3
**Presentation:** 4
**Contribution:** 3
**Rating:** 8
**Confidence:** 3

**Summary:**

The paper introduces a multi-physics benchmark dataset.

The paper describes $6$ paired tasks and evaluates the performance with $4$ models.

The paper also demonstrates that the current models' performance may suffer when naively applying the method to this dataset and proposes two strategies to alleviate the problem.

The paper provides data and the script to generate the data, and the baseline code (at least planned).

**Strengths:**

A new problem that has practical implications for modeling multi-physics systems.

**Weaknesses:**

It may not be of general interest.

**Questions:**

A fundamental question is if this class of problems can be reduced to a single PDE and the conditions under which it is possible to do this.

Another interesting question is that if we had a unique PDE, what would make a problem more difficult to solve?

---

> ### Author Response · Authors · 2025-11-23
> **Response to Reviewer 6qbz**
>
> Thank you to the reviewers for your recognition of our work! Below are the responses to the questions you raised.
> ***
> **W1:** It may not be of general interest.
>
> **A1:** Multiphysics problems have broad and significant engineering relevance, with critical applications in semiconductor manufacturing, aerospace, biomedicine, geophysics, and precision instrumentation. The applicability in real scenarios is also an important pursuit of the SciML community. Addressing multiphysics scenarios is therefore a necessary step toward bridging the gap between SciML research and practical deployment.
> ***
> **Q1:** A fundamental question is if this class of problems can be reduced to a single PDE and the conditions under which it is possible to do this.
>
> **A2:** This question can be addressed from two complementary perspectives.
> First, a multiphysics system typically involves at least three coupled PDEs. Reducing these equations into a single PDE would significantly increase the order and structural complexity of the resulting equation, making it extremely difficult to solve in practice. Moreover, the original equations often describe different physical fields that may be scalar, vector, or tensor quantities. Compressing them into a single PDE obscures these structural distinctions and further increases mathematical and computational difficulty. When using SciML-based methods, collapsing multiple PDEs into a single equation may also result in a substantial loss of physical information encoded in the original system.
> Second, if the problem is simplified during modeling, a coupled system can sometimes be approximated by a single-physics PDE. Many existing studies adopt this approach. For example, in an electro-thermal coupling problem, if one assumes that the electric field is independent of the temperature field and treats it as a fixed function or constant, then the temperature equation can be solved independently. However, doing so introduces non-negligible errors. In our electro-thermal case, the primary concern is chip temperature in design. The electromagnetic properties of the chip are highly sensitive to temperature. When fully accounting for electro-thermal coupling, the peak temperature rise is 11.84 K, whereas solving only the thermal equation with a fixed electric field yields 9.15 K, resulting in a 29.4% relative error. This discrepancy arises from neglecting the temperature dependence of electromagnetic parameters.
> Therefore, if an application can tolerate ~30% error, or if the electromagnetic parameters are known to be insensitive to temperature, the system may indeed be simplified into a single-physics PDE. Otherwise, such a reduction is inappropriate and would lead to misleading conclusions.
> ***
> **Q2:** Another interesting question is that if we had a unique PDE, what would make a problem more difficult to solve?
>
> **A3:** If the problem is already formulated or reduced to a single PDE, several factors may still make it difficult to solve:
>
> **(1) High-order PDE difficulties in SciML methods.**
> For SciML models, high-order differential operators introduce significant challenges. Neural networks and automatic differentiation often struggle to compute high-order derivatives with sufficient accuracy and numerical stability, leading to training difficulties. Strong nonlinear terms further increase optimization complexity and may trap the model in local minima, preventing it from effectively approximating the full solution space.
>
> **(2) Spatial multiscale characteristics.**
> Many practical problems exhibit large variations in spatial scales. For example, in chip-level simulations, certain structural features are on the micrometer scale, whereas PCB-level structures are on the millimeter scale. Using a uniform micrometer-level resolution would result in prohibitive memory and computation costs. Designing a method that can efficiently handle such multiscale spatial variations remains a major challenge.
>
> **(3) Material heterogeneity and anisotropy.**
> PDEs may involve material properties that vary spatially or exhibit anisotropic behavior. This makes the input features highly non-stationary, increasing the difficulty for learning a generalizable mapping. As a result, the model becomes harder to train and the learned patterns may not generalize well across different regions or configurations.
>
> We have added some descriptions about single-physics and multi-physics in the **Multiphysics Problems** section of the main text.
> ***
> We hope our responses have helped clarify your concerns and welcome further discussion!

---

### Official Review · Reviewer_izj7 · 2025-10-31

**Soundness:** 3
**Presentation:** 3
**Contribution:** 4
**Rating:** 6
**Confidence:** 4

**Summary:**

This paper introduces a novel benchmark (Multiphysics Bench) for evaluating Scientific Machine Learning methods applied to coupled multiphysics Partial Differential Equations (PDEs), consisting of six canonical coupling problems that cover diverse scenarios like bidirectional/unidirectional coupling and both steady-state/frequency-domain and transient dynamics. A robust baseline is established by rigorously testing four machine learning approaches: Physics-Informed Neural Networks (PINNs), Fourier Neural Operators (FNO), DeepONet, and DiffusionPDE.

**Strengths:**

1. This paper systematically focuses on the critical and complex domain of coupled multiphysics PDEs. The inclusion of diverse and challenging coupling types (bidirectional/unidirectional, equation/parameter-level) demonstrates an original and comprehensive problem formulation that accurately reflects real-world engineering and science.
2. The authors use four leading learning-based PDE solvers (PINN, FNO, DeepONet, and DiffusionPDE) on all six problems
3. The problems are deliberately chosen to expose specific technical weaknesses of current models, such as gradient inconsistencies in PINNs and mode collapse in FNOs when applied to coupled systems.
4. The generation of the dataset, particularly its size ($10^4$ training samples) and the complexity of the output fields (e.g., 12 output channels for Acoustic-Structure coupling), represents a significant benchmark for training modern learning-based PDE solvers.

**Weaknesses:**

1. The paper needs to explain why the relative $L_2$ error performance of baseline models remains unchanged (or nearly unchanged) despite a significant increase in the data scale (number of training samples) in Table 4.
2. The paper correctly identifies various failure modes (e.g., FNO mode collapse, PINN gradient inconsistency), but the justification is often descriptive rather than quantitatively analytical.

**Questions:**

See the weaknesses.

---

> ### Author Response · Authors · 2025-11-23
> **Response to Reviewer izj7**
>
> We sincerely thank the reviewer for the constructive and insightful comments. We appreciate your recognition of our work on multiphysics problems, as well as your positive feedback on the quality of our reporting. Below, we provide responses to your main concerns.
> ***
> **W1:** The paper needs to explain why the relative  error performance of baseline models remains unchanged (or nearly unchanged) despite a significant increase in the data scale (number of training samples) in Table 4.
>
> **A1:** We appreciate the reviewer’s observation. As discussed in the **Data Scaling** part in our paper, current SciML models exhibit limited scaling capability in multiphysics settings.
> The current SciML framework often reaches a saturation state after approximately 10,000 samples, making it difficult to extract effective information from larger datasets. Moreover, due to weak scalability across coupled physical variables, these models struggle to capture cross-field dynamic interactions. As a result, increasing the dataset size does not yield noticeable improvements in generalization performance.
> ***
> **W2:** The paper correctly identifies various failure modes (e.g., FNO mode collapse, PINN gradient inconsistency), but the justification is often descriptive rather than quantitatively analytical.
>
> **A2:**
> We observe that FNO produces nearly identical output fields under different input conditions, indicating that the network fails to generate distinguishable outputs for inputs that are “similar but not identical.” This behavior is clearly illustrated in the FNO rows of Fig. 15–20 in the revised paper. In addition, Fig. 23 provides a direct input–output comparison for two different cases. The FNO prediction results in the "diff" row shown in Fig. 23 is almost zero-valued, demonstrating that FNO is essentially unresponsive to input perturbations. The magnitude of the predicted differences is far smaller than the true variation scale, which is a typical signature of mode collapse.
>
> For the gradient inconsistency issue in PINNs, we similarly add quantitative analysis. As shown in Fig. 21 and Fig. 22 of the revised paper (Electro–Thermal Coupling and Thermo–Fluid Coupling), the PDE loss of the electric-field equation is one to two orders of magnitude larger than that of the temperature equation, and the temperature PDE loss remains nearly constant throughout training. This arises because the electric field exhibits rapid spatial variations and large gradients, while the temperature field varies much more slowly. Consequently, during joint optimization, the PINN prioritizes fitting the electric-field equation, suppressing the gradients associated with the temperature equation and leading to a characteristic gradient competition phenomenon in multiphysics learning.
> ***
> We hope that the above responses can address the concerns of the reviewers. If there are any further questions, we are more than happy to have further communication with you!

---

### Official Review · Reviewer_fhMJ · 2025-11-02

**Soundness:** 2
**Presentation:** 3
**Contribution:** 2
**Rating:** 2
**Confidence:** 4

**Summary:**

This paper presents Multiphysics Bench, a dataset and benchmark suite for evaluating SciML PDE solvers on multiphysics problems. It consists FEM-generated data for a collection of six multiphysics problem (electro-thermal, thermo-fluid, electro-fluid, magneto-hydrodynamic, acoustic–structure, mass-transport–fluid). The paper also evaluates four classes of SciML solvers (PINNs, FNO, DeepONet, DiffusionPDE), reports empirical findings, and suggests a couple of "tricks" to help improve performance.

While the paper presents a solid engineering effort, it offers limited scientific insights or conceptual contribution. It finds that SciML tools behave the same way on larger, coupled PDEs, which is interesting to note, but it neither advances our understanding of the unique challenges presented by multiphysics over single-physics problems for SciML methods, nor offer any deeper insights into the generalization of SciML methods.

**Strengths:**

- Multi physics problems are widely present in the real-world.

highly Creating a standardized, multi-scenario multiphysics benchmark is valuable for the SciML community; the chosen scenarios are relevant to real applications (electronics, fluid/thermal systems, acoustics, porous-media transport).
- Many different PDEs are included in the benchmark with different initial and boundary conditions, although there are a few omissions.
- Evaluation of four major solver families (PINNs, DeepONet, FNO, DiffusionPDE), although there are a few omissions. The paper also reports many metrics (RMSE, relative L2, MaxError, etc.)
- The paper alludes to practical issues such as imbalance of residual magnitudes and degradation in transient tasks and suggests remedies like quantile normalization and auto-balanced weighting.

**Weaknesses:**

Major Weakness: The scientific contribution of the paper is unclear.

- The multiphysics benchmark dataset does not focus on what sets multiphysics problems apart from single-physics problems (cross-field interactions, different spatio/temporal scales across fields, etc.). Simply measuring global reconstruction error does not inform us why SciML methods might fail on multiphysics problems or how to fix them?

- The empirical takeaway is essentially that existing SciML models for PDEs generalize to multiphysics problems about as well as they do for single physics problems. This is neither surprising not theoretically illuminating. In fact, it suggests that the multiphysics coupling is weak or moderate. So, the benchmark does not isolate multiphysics difficulty.

- The benchmark currently emphasizes steady-state (frequency-domain) systems, and has only one transient case. So, the temporal diversity is limited and excludes a class of time-dependent or dynamically unstable multiphysics systems that are highly relevant in practice. So, the benchmark does not fully capture the challenges of transient coupling, time-scale stiffness, or nonlinear feedback dynamics, which are common in many real-world multiphysics problems.

Other Weaknesses:

- SciML methods are known to be very brittle and sensitive to hyperparameters (see [1]). The empirical results in the paper may not be reliable. There are no confidence intervals in the majority of the results (except Figure 9).

- The abstract and the introduction point to insights and "bag of tricks". These insights do not appear to be different from single physics scenarios, and the "bag of tricks" are not substantial enough to be included in the main paper.

- The benchmarks do not capture real-world operational challenges, where measurements are noisy, PDE parameters are not known exactly, PDEs are approximations of the underlying physical phenomenon. The utility of evaluating on a benchmark that does not incorporate the such real-world challenges is unclear.

[1] McGreivy, Nick, and Ammar Hakim. "Weak baselines and reporting biases lead to overoptimism in machine learning for fluid-related partial differential equations." Nature Machine Intelligence 6, no. 10 (2024): 1256-1269.

**Questions:**

- How sensitive are the SciML methods to hyperparameter tuning? Would deeper FNOs, or PINNs with different loss weights change the conclusions?

- Do any of the SciML methods violate conservation laws in multiphysics problems? If yes, can the violation be quantified?

- Do PINN residuals overfit to one physics and underfit to another?

- How would foundation models for PDEs like Poseidon [2] perform?

- There is a claim in the introduction: "By aligning computational solvers more closely with the governing physical laws, MultiphysicsBench provides a robust foundation for developing models that are more accurate, resilient, and generalizable—paving the way for future advances in simulating complex, coupled physical systems." This is not apparent to the reviewer. Can you elaborate how the contributions of the paper aligns with this claim?

[2] Poseidon: Efficient Foundation Models for PDEs, NeurIPS 2024

---

> ### Author Response · Authors · 2025-11-23
> **Response to Reviewer fhMJ – Part 1**
>
> We sincerely appreciate the reviewer's detailed and insightful comments.
> ***
> **Summary:** While the paper presents a solid engineering effort, it offers limited scientific insights or conceptual contribution. It finds that SciML tools behave the same way on larger, coupled PDEs, which is interesting to note, but it neither advances our understanding of the unique challenges presented by multiphysics over single-physics problems for SciML methods, nor offer any deeper insights into the generalization of SciML methods.
>
> **A0:** We sincerely thank the reviewer for the thoughtful feedback. Below we clarify the scientific contributions of our work.
>
> **(1) First benchmark dataset for coupled multiphysics PDEs.**
> The existing scientific computing and modeling benchmark tests mainly focus on a single physical process or simple simulation problems. In contrast, our dataset is the first to comprehensively cover real multiphysics field coupling, including variations in geometry, material properties, boundary conditions, and initial conditions. This enables a rigorous evaluation of SciML behavior under realistic coupled systems.
>
> **(2) New empirical finding: SciML does not generalize from single physics to multiphysics.**
> Our evaluation of four representative SciML architectures reveals a key scientific finding: models that perform extremely well on single-physics PDEs (e.g., FNO achieving 0.86% error on Navier–Stokes[1]) fail on coupled systems (e.g., 43.53% error on Thermal–Navier–Stokes). This exposes a clear limitations of generalization phenomenon that has not been studied in prior work.
>
> **(3) Why existing methods fail: multiphysics introduces previously unexplored challenges.**
> Strong multiphysics scale disparity introduces difficulties. For example, in electro–thermal coupling, different fields can differ by up to $10^4$ in numerical magnitude, and spatial smoothness varies significantly across fields. Our geometries, boundary conditions, and material parameters are also derived from real engineering configurations rather than idealized academic setups. Discontinuities and singular behaviors further complicate learning; for instance, in the Elder problem at $t=0$, concentration and velocity exhibit small-scale singularities that break typical normalization strategies used in SciML training. To address these issues, we propose several stabilization and preprocessing techniques that enable current SciML models to run on complex multiphysics datasets. These techniques can serve as initial tools for researchers developing next-generation multiphysics-aware SciML models.
> In addition, multiphysics coupling leads to a large number of input/output channels and intricate interactions among fields, making existing SciML architectures insufficient for solving coupled PDEs. We wwill design new SciML architectures specifically tailored for strongly coupled multiphysics systems in our future work.
>
> Further details are given in the subsequent response. We hope this clarification resolves the reviewer’s concern.
> ***
> The scientific contributions of our work are summarized above. We now address the weaknesses raised by the reviewer.
>
> **W1:** The multiphysics benchmark dataset does not focus on what sets multiphysics problems apart from single-physics problems (cross-field interactions, different spatio/temporal scales across fields, etc.). Simply measuring global reconstruction error does not inform us why SciML methods might fail on multiphysics problems or how to fix them?
>
> **A1:** Thank you for highlighting this important point. We explicitly designed our multiphysics dataset to include cross-field interactions and different spatio-temporal scales across fields. The cross-field couplings are inherently included in the PDEs solved during data generation, as indicated in Table 1 (coupling type) and Equations (5)–(10), with detailed explanations provided in Appendix A.1. In meaningful multiphysics problems, the equations for each field are not solved independently, instead, each field is constrained and influenced by other fields.
>
> In our designed physical systems, the coupled PDEs indeed involve multiple forms of multiscale behavior, including differences in spatial scales, temporal scales, and magnitude scales across fields. For example, in Electro–thermal Coupling, the electric field varies sharply in space, whereas temperature diffusion evolves over a much larger spatial volume. In Magneto–Hydrodynamic (MHD) Coupling, current density can increase rapidly near boundaries due to the skin effect, while the velocity field may develop localized vortices that exhibit high-frequency structures. In Acoustic–Structure Coupling, acoustic wavelengths are typically on the meter scale, whereas structural deformation occurs on the millimeter scale, and these numerical-scale differences can be directly observed in Figures 15–20 in revised paper.

---

> ### Author Response · Authors · 2025-11-23
> **Response to Reviewer fhMJ – Part 2**
>
> **Continue with A1:** In the Mass Transport–Fluid Coupling time-dependent problem, the timescale of advection is much faster than that of diffusion; diffusion can be slower by a factor of $10^4–10^5$. As a result, shortly after a substance is injected, advection rapidly reshapes the local concentration distribution, while diffusion evolves extremely slowly. This produces fast, localized spatiotemporal variations in the concentration field, whereas the flow field changes more smoothly, which is a hallmark of multiscale, multiphysics coupling.
> Overall, these characteristics pose significant challenges for SciML: the large disparities in temporal, spatial, and numerical scales, together with strong PDE coupling, lead to highly stiff systems. This requires the model to simultaneously capture both high-frequency and low-frequency behaviors, and the large number of physical fields further increases the input–output channel dimensionality, making scientific machine learning considerably more difficult. Furthermore, we have added some descriptions in the **Multiphysics Problems** section of the main text.
> ***
> **W2:** The empirical takeaway is essentially that existing SciML models for PDEs generalize to multiphysics problems about as well as they do for single physics problems. This is neither surprising not theoretically illuminating. In fact, it suggests that the multiphysics coupling is weak or moderate. So, the benchmark does not isolate multiphysics difficulty.
>
> **A2:** Our evaluation of four representative SciML architectures reveals a key scientific finding: models that perform extremely well on single-physics PDEs (e.g., FNO achieving 0.86% error on Navier–Stokes [1]) fail on coupled systems (e.g., 43.53% error on Thermal–Navier–Stokes). This performance gap indicates that multiphysics coupling introduces nontrivial challenges for existing architectures, contradicting the assumption that current SciML methods generalize equally well to singlephysics and multiphysics regimes.
> Importantly, multiphysics interactions are not optional in real applications. In many engineering systems, the governing fields are intrinsically coupled and cannot be solved independently. For instance, in electro-thermal analysis of integrated circuits, the temperature distribution cannot be obtained by solving the heat equation alone because the heat sources are generated by electrical conduction. Accurate prediction therefore requires solving the coupled electro-thermal PDEs, where one field directly determines the forcing term of the other. This illustrates why isolating single-physics problems does not reflect actual application difficulty, and why benchmarks must explicitly evaluate coupled PDE systems.
> Our benchmark is designed precisely to expose this gap. The substantial performance deterioration observed across models demonstrates that multiphysics coupling presents a genuine and previously under-explored challenge.
> ***
> **W3:** The benchmark currently emphasizes steady-state (frequency-domain) systems, and has only one transient case. So, the temporal diversity is limited and excludes a class of time-dependent or dynamically unstable multiphysics systems that are highly relevant in practice. So, the benchmark does not fully capture the challenges of transient coupling, time-scale stiffness, or nonlinear feedback dynamics, which are common in many real-world multiphysics problems.
>
> **A3:** In response to your suggestion, we have incorporated a transient fluid–solid thermal coupling example to simulate the deformation behavior of bimetallic wings subjected to aerodynamic heating in spacecraft applications. This bidirectionally coupled multiphysics scenario involves high-speed airflow inducing thermal loads on the metallic wing structure, which subsequently undergoes deformation. The resulting structural changes further influence the surrounding flow field, creating a complex interaction between thermal, fluid, and structural dynamics.
> The dataset comprises 10,000 samples encompassing wing geometries, heat source distributions, flow fields, deformation fields, and temperature fields. We believe this dataset significantly contributes to advancing research in aerospace engineering by providing a comprehensive resource for studying thermally induced structural-fluid interactions. The dataset has been fully generated and preliminary evaluated using the DeepONet architecture. The corresponding results are summarized as follows:
> | Model    | $T$         | $u$         | $v$         | $x$         | $y$         |
> |----------|--------------|--------------|--------------|--------------|--------------|
> | **DeepONet** | 2.032×10⁻⁴  | 9.474×10⁻²  | 6.010×10⁻²  | 5.078×10⁻⁴  | 3.984×10⁻⁴  |
>
> where $T$ denotes temperature，$u$ and $v$ represent the fluid velocities in the $x$-direction and $y$-direction, respectively. And
> $x$ and $y$ denote the solid displacements in the $x$-direction and $y$-direction.

---

> ### Author Response · Authors · 2025-11-23
> **Response to Reviewer fhMJ – Part 3**
>
> **Continue with A3:** In addition, there is another newly added time-domain Schrödinger-Poisson self-consistent equation. The descriptions of the two new time-domain problem can be find in Appendix A.1 of the article.
> ***
> **W4:** SciML methods are known to be very brittle and sensitive to hyperparameters (see [1]). The empirical results in the paper may not be reliable. There are no confidence intervals in the majority of the results (except Figure 9).
>
> **A4:** In all experiments, the reported relative errors are computed as the average over 1,000 test samples. In addition, we provide the error distributions and corresponding confidence intervals for four multiphysics problems  as shown in Fig11-Fig14 in our revised paper. The remaining two problems involve a larger number of coupled physical fields, so their distribution plots are not shown in the text. We will include these results in a subsequent update.
>
> Regarding hyperparameter selection, we strictly adhered to a fairness principle. To ensure a fair comparison, we chose hyperparameters such that the model scale (i.e., the number of trainable parameters) remains approximately consistent across all baselines, as described in Section 4 (Baseline Setup) and Appendix B.2 of the paper.
> ***
> **W5:** The abstract and the introduction point to insights and "bag of tricks". These insights do not appear to be different from single physics scenarios, and the "bag of tricks" are not substantial enough to be included in the main paper.
>
> **A5:** Strong multiphysics scale disparity introduces difficulties. For example, in electro–thermal coupling, different fields can differ by up to $10^4$ in numerical magnitude, and spatial smoothness varies significantly across fields. Our geometries, boundary conditions, and material parameters are also derived from real engineering configurations rather than idealized academic setups. Discontinuities and singular behaviors further complicate learning; for instance, in the Elder problem at $t=0$, concentration and velocity exhibit small-scale singularities that break typical normalization strategies used in SciML training.
> To make existing SciML baselines even trainable on such data, we introduce stabilization techniques that are not needed in single-physics settings.
>
> (1) Adaptive PDE-loss balancing, which dynamically rescales losses across fields of different magnitudes, while fixed weights used in singlephysics models lead to divergence in multiphysics training.
>
> (2) Quantile normalization, which remains stable under discontinuities and singular structures, unlike conventional min–max normalization.
>
> These are necessary methodological adaptations that enable existing models to operate on multiphysics systems. They also reveal limitations of current SciML methods when confronted with coupled PDEs. In future work, we will extend these initial solutions toward dedicated multiphysics-aware training strategies and architectures.
> ***
> **W6:** The benchmarks do not capture real-world operational challenges, where measurements are noisy, PDE parameters are not known exactly, PDEs are approximations of the underlying physical phenomenon. The utility of evaluating on a benchmark that does not incorporate the such real-world challenges is unclear.
>
> **A6:** Thank you for your valuable suggestions.In other PDE dataset studies, most do not consider situations where parameters are unknown and measurements are noisy, such as PDE Bench [2].
> For the situation where parameters are unknown, traditional numerical methods cannot solve it anymore. This highlights the advantages of SciML. However, at this time, the two methods in our baseline (FNO and DeepOnet) are actually not affected by anything. They only need to use the input and output images to solve the problem, and do not need equation information during the solution process. For PINN and DiffusionPDE methods, a pde loss term is added in the solution process, but there is also a data loss part. Therefore, certain simulation results can still be obtained. As shown in tables 3 and 21 in the article. If we input incomplete physical equation information (the incomplete part), we can still obtain certain simulation results, but the errors are relatively large, around 40%, indicating that this part of multi-physics information is not well reflected at present.
> For the situation where measurements are noisy, in addition, we added a set of noisy cases to our data set. We tested the Electro–Thermal situation. The results are shown in table 20 in the appendix.
>
> ***
> [1] Zongyi Li, Nikola Kovachki, et al. "Fourier Neural Operator for Parameter Partial Differential Equations." ICLR, 2019.
>
> [2] M. Takamoto, et. al, “PDEBench: an extensive benchmark for scientific machine learning,” NeurIPS 2022.

---

> ### Author Response · Authors · 2025-11-23
> **Response to Reviewer fhMJ – Part 4**
>
> **Q1:** How sensitive are the SciML methods to hyperparameter tuning? Would deeper FNOs, or PINNs with different loss weights change the conclusions?
>
> **A7:** To ensure a fair comparison, we selected hyperparameters to keep the model scale (i.e., number of trainable parameters) approximately consistent across baselines. Based on this, we conducted grid searches for key hyperparameters such as learning rate and training epochs to optimize performance.
> Within the range of all reasonable hyperparameters, the main conclusions of the paper remain unchanged. Although the absolute error may have minor variations, the reported comparative trend in the paper is stable. A deeper FNO structure or different PINN loss weights do not fundamentally alter the results. Changing the depth of FNO and the parameters of PINN cannot solve the difficulties currently encountered by the SciML methods in solving complex PDE equations.
>
> The fundamental reason why the PINN method fails in solving complex PDE problems in multi-physics fields is that when the method involves high-order derivative terms in the equation, differentiation will generate noise. And when dealing with rigid PDE structures, it is difficult to solve. It is difficult to capture high-frequency information, and in multiphysics problems, it will face the problem of multiple PDE loss conflicts. Once entering high-dimensional, coupled, multi-scale, and rigid PDE, the optimization and representation capabilities of PINN quickly reach their limits. The performance of the FNO method in frequency-domain problems is quite ordinary because FNO tends to learn the low-frequency part. When the differences between the inputs are only small-scale local variations, the low-frequency bias of FNO will ignore these differences, and the output will become very smooth, resulting in mode collapse.
> ***
> **Q2:** Do any of the SciML methods violate conservation laws in multiphysics problems? If yes, can the violation be quantified?
>
> **A8:** Methods such as PINN and DiffusionPDE, which incorporate PDE loss during training or inference, naturally satisfy conservation laws. In contrast, methods like FNO and DeepONet, which merely learn input–output mappings, may violate conservation laws. We can quantify conservation-law violations using the PDE residuals:
>
> $\varepsilon_{\text{PDE},i} = \|\|  N_i(u_\theta) - f_i \|\|_{L^2(\Omega)}, \quad i = 1,2,\dots$
>
> where $N_i(u_\theta)$ is the PDE operator for a coupled physical field, $f_i$ is the corresponding source term, $u_θ$ is the predicted field, and $Ω$ denotes the spatial domain.
> ***
> **Q3:** Do PINN residuals overfit to one physics and underfit to another?
>
> **A9:**  In multiphysics PINNs, it is indeed common for one PDE residual to be well-fitted by the model while another residual barely decreases. This imbalance arises mainly from differences in physical scales, frequencies, and gradient magnitudes among the PDEs. For example, in the electro-thermal coupling problem of Equation (5), the electric field varies rapidly in space and has high frequency, leading to large second-order derivatives and hence large residual gradients. In contrast, the temperature field changes more slowly and smoothly, resulting in smaller residual gradients. Consequently, the model may overfit one field while underfitting the other. Moreover, since the source term of the heat equation depends on the square of the electric field, the temperature field contributes less to the heat PDE loss, making its optimization even more difficult.
>
> We present the training losses of the FNO model for the Electro-Thermal task and the Thermo-Fluid task. In the revised manuscript, Figures 21 and 22 show that the magnitudes of the losses for different partial differential equations differ by tens to even hundreds of times. To mitigate this issue, we employ an adaptive weighting strategy that dynamically adjusts the loss weights of different PDEs during training, which can partially alleviate underfitting in certain fields.

---

> ### Author Response · Authors · 2025-11-23
> **Response to Reviewer fhMJ – Part 5**
>
> **Q4:** How would foundation models for PDEs like Poseidon [2] perform?
>
> **A10:** Following your suggestion, we further extended the evaluation scope by including Poseidon, a foundation model based on the operator transformer. This model leverages a multi-scale operator transformer to learn PDE solution operators, treating a PDE as an operator that maps input fields to solution fields. During pretraining, the model is trained on PDE data from fluid dynamics and subsequently fine-tuned on other types of PDE problems.
> As a case study, we applied Poseidon to Electro-Thermal Coupling, Thermo-Fluid Coupling, and Electro-Fluid Coupling problems. The mean squared errors (corresponding relative errors) on the test dataset are summarized as follows:
> |    Electro-Thermal    | **Re($E_z$)**        | **Im($E_z$)**        |   **$T$**   |
> |--------|-----------|-----------|-----------|
> | **Poseidon**    | $4.248×10^{-1}$ | $4.318×10^{-1}$ | $1.465×10^{-3}$        |
>
> |    Thermo-Fluid    |   **$u$**        |   **$v$**        |   **$T$**   |
> |--------|-----------|-----------|-----------|
> | **Poseidon**    | $4.568×10^{-1}$ | $4.026×10^{-1}$ | $1.365×10^{-4}$        |
>
> |    Electro-Fluid    |   **$V$**        |   **$u$**        |   **$v$**    |
> |--------|-----------|-----------|-----------|
> | **Poseidon**    | $2.292×10^{-1}$ | $1.989×10^{-1}$ | $8.437×10^{-1}$        |
>
> Due to the limited time, we tested three physics problems. The results show that the effect is similar to those of other existing SciML models, but there is still much room for improvement in performance. This highlights the necessity of designing methods specifically for coupled multi-physics problems.
> ***
> **Q5:** There is a claim in the introduction: "By aligning computational solvers more closely with the governing physical laws, MultiphysicsBench provides a robust foundation for developing models that are more accurate, resilient, and generalizable—paving the way for future advances in simulating complex, coupled physical systems." This is not apparent to the reviewer. Can you elaborate how the contributions of the paper aligns with this claim?
>
> **A11:** Multifield simulation is closer to real physical problems than single-field simulation. Multifield simulation is not merely the simple superposition of multiple PDEs, but involves the coupling relationships between different physical fields, which cannot be reflected in single-field simulation. We have explicitly considered these coupling relationships in the dataset design, so the simulation results are more in line with the physical world.
> When generating multifield datasets, the parameters and initial/boundary conditions of each physical field must follow the actual physical laws, otherwise, correct multi-field coupling cannot be achieved. For example, in the electro-thermal coupling problem, the input current will generate an electric field in the chip, which will further cause the temperature of the structure to rise. To obtain a reasonable temperature distribution, the current, electric field, and material parameters must conform to the real physics, rather than being just simplified toy cases.
> Therefore, multifield simulation itself requires the model and data to be more closely aligned with the actual physical problem. Our benchmark covers various types of multifield coupling, providing a real platform for subsequent researchers to test and develop SciML methods, thereby promoting the development of machine learning methods for multifield simulation. MultiphysicsBench lays a solid foundation for the development of future SciML models for simulating complex coupled multiphysics PDEs.
> ***
> We hope that these additional experiments and explanations address the reviewer’s concerns, and we respectfully invite you to reconsider our submission.

---

### Author Response · Authors · 2025-12-02
**Summary of Revisions and Responses to Reviews**

Dear Program Chairs, Senior Area Chairs, and Area Chairs,

With the conclusion of the discussion phase, we would like to briefly summarize the current review status and the key efforts we have made during the rebuttal process. We have carefully addressed all the reviewers’ concerns, and no new issues were raised in the discussion.
***
**Our major revisions and efforts are summarized as follows:**
- Regarding the reviewers’ concerns about the distinction between single-physics and multiphysics problems, we have systematically clarified the fundamental differences in modeling complexity and numerical solution difficulty in both the manuscript and the response. In particular, our method explicitly accounts for the complex and strongly coupled interactions among multiphysics problems, which have not been adequately considered in prior studies. In addition, the baseline results clearly demonstrate that existing SciML solvers exhibit limited performance on multiphysics problems. These points have also been further strengthened in the revised manuscript.
- To address the concern regarding the limited number of time-dependent cases, we added two new transient examples: (1) a transient fluid–structure–thermal coupling example that simulates the deformation of a bimetallic spacecraft wing under aerodynamic heating, and (2) a time-domain self-consistent Schrödinger–Poisson system, which supports the simulation of modern electronic and optoelectronic devices.
- In response to the request for baseline models under a multiphysics framework, we introduced a newly emerging framework, M2PDE, and conducted additional experiments on the electro-thermal dataset.
- We also evaluated the Poseidon foundation model, as suggested by the reviewers. The results show that its performance is comparable to other baseline models under the same limited numerical precision, which further highlights the necessity and significance of research specifically tailored for multiphysics problems.
- In addition, guided by the reviewers’ valuable suggestions, we included more experimental results, figures, and textual explanations, such as noise robustness experiments, to further strengthen the completeness and clarity of the paper.
***
**Regarding the reviewers’ feedback:**
- Reviewer **fhMJ** raised several detailed concerns, all of which have been thoroughly addressed in our response. Some of the concerns were based on a partial misunderstanding of our contributions, whereas our work in fact directly targets the key issues raised. The revised manuscript makes these contributions more explicit. The reviewer did not raise any further questions after our response, and we believe that most of the original concerns have been effectively resolved.
- Reviewer **izj7** expressed strong recognition of our work, noting that it is original and comprehensive, covering diverse and challenging coupling types, and that it accurately reflects real-world engineering and scientific settings.
- Reviewer **6qbz** gave highly positive feedback, considering our work a meaningful and practically valuable new problem formulation for multiphysics system modeling.
- Reviewer **rneb** also acknowledged the contributions of our work, confirmed that his or her concerns had been addressed in our response, and expressed satisfaction with the revisions.
***
Through these revisions, we believe that the overall quality and completeness of our manuscript have been significantly improved.

We sincerely seek your kind consideration of our rebuttal and the revised manuscript to assist in your final decision.
We greatly appreciate your valuable time, effort, and thoughtful evaluation throughout this process.

Best regards,

The Authors

---

### Meta-Review · Area_Chair_tzJe · 2026-01-10

**Summary:**

Below is the summary of the concerns of the reviewers
fhMJ:
1) The scientific contribution of the paper is unclear: the multiphysics benchmark dataset does not focus on what sets multiphysics problems apart from single-physics problems (cross-field interactions, different spatio/temporal scales across fields, etc.). Existing methods generalize and multiphysics coupling is weak.
2) Mostly steady-state systems, only one transient case.
izj7:
1) Table 4: why relative error performance of baseline models remains unchanged despite a significant increase in the data scale.
2) The paper correctly identifies various failure modes (e.g., FNO mode collapse, PINN gradient inconsistency), but the justification is often descriptive rather than quantitatively analytical.
6qbz
Only one and strange: "It may not be of general interest"
rneb
1) Stronger baselines for coupled PDEs
2) The main contribution is the new dataset, but there are no new solvers tailored to the specific challenges these datasets present.
3) The citation style is incorrect for ICLR guidelines (\citep should be used for parenthetical citations).

**Reviewer Concerns:**

The most detailed review has been given by fhMJ, and the authors have provided a comprehensive rebuttal.
However, I don't agree that the rebuttal answers the concern, it also contains very strange statements, for example 'PINNs satisfy conservation laws' which are simply incorrect. So I think the concerns of this reviewer have not been addressed. The concerns of the other reviewers have been addressed.

**Reviewer Scores:**

The scores are 2, 6, 8, 6 and although 3 reviewers are positive, they did not provide a comprehensive review, so I think the the negative reviewer is correct in this case.

---

### Decision · Program_Chairs · 2026-01-26

Reject